# When LLMs Encounter Open-world Graph Learning: A Fresh View on Unlabeled Data Uncertainty

## Abstract

Recently, large language models (LLMs) have driven a systematic shift in the graph ML community through the adoption of text-attributed graphs (TAGs). Although a variety of frameworks have been developed, most fail to properly address the challenge of data uncertainty in open-world environments, which is vital for real-world deployment. A representative source of such uncertainty is the limited availability of labels in large-scale datasets due to high annotation costs, where unlabeled nodes may either belong to known classes or represent novel, unknown classes. While node-level out-of-distribution detection and conventional open-world graph learning attempt to tackle this problem, two core limitations remain: Insufficient methods — TAGs integrate textual and structural information, yet existing approaches typically optimize semantics or topology in isolation for unknown-class rejection, limiting their effectiveness; ② Incomplete pipelines — handling unknown-class nodes is essential for model re-updates and long-term deployment, but most studies conduct only idealized analyses, such as assuming a predefined number of unknown classes, which restricts practical utility. To overcome these issues, we introduce the Open-world Graph Assistant (OGA), an LLM-based framework. OGA first performs unknown-class rejection via adaptive label traceability (ALT), harmoniously combining semantic and topological cues, and then applies the graph label annotator (GLA) for unknown-class annotation, allowing unlabeled nodes to contribute to model training. In essence, OGA offers a new pipeline that fully automates the handling of unlabeled nodes in open-world environments, and we establish a systematic benchmark covering four key aspects to validate its effectiveness and practicality through extensive experiments.

## 1 Introduction

In recent years, graph neural networks (GNNs) have emerged as a pivotal technology for modeling relational data, enabling the generation of high-quality embeddings by simultaneously encoding both feature and structural information Besta et al. (2022); Kipf & Welling (2017); Xu et al. (2019). This capability bridges diverse application scenarios and graph-based downstream tasks, enhancing the practical significance of GNNs in the real world.

In the era of large language models (LLMs), this graph ML paradigm is experiencing accelerated advancements, driven by the emergence of text-attributed graphs (TAGs), where nodes and edges are equipped with textual information. This graph-text data integration has facilitated the collection of metadata and the development of numerous frameworks. However, they often struggle to address the uncertainty in rapidly expanding metadata. Given the complexity of this data uncertainty issue, in this paper, we particularly focus on the under-labeling problem in TAGs due to costly labor.

During our investigation, we found that node-level out-of-distribution (OOD) detection and conventional open-world graph learning align most closely with the context of our research problem. Additionally, we identify that related fields such as continual, incremental, and lifelong graph learning are also relevant to our research Qi et al. (2025); Kou et al. (2020); Lin et al. (2023); Niu et al. (2024); Choi et al. (2024); Zhou & Cao (2021); Hoang et al. (2023); Zhang et al. (2022). However, these fields primarily focus on streaming data management under uncertainty, which differs from our

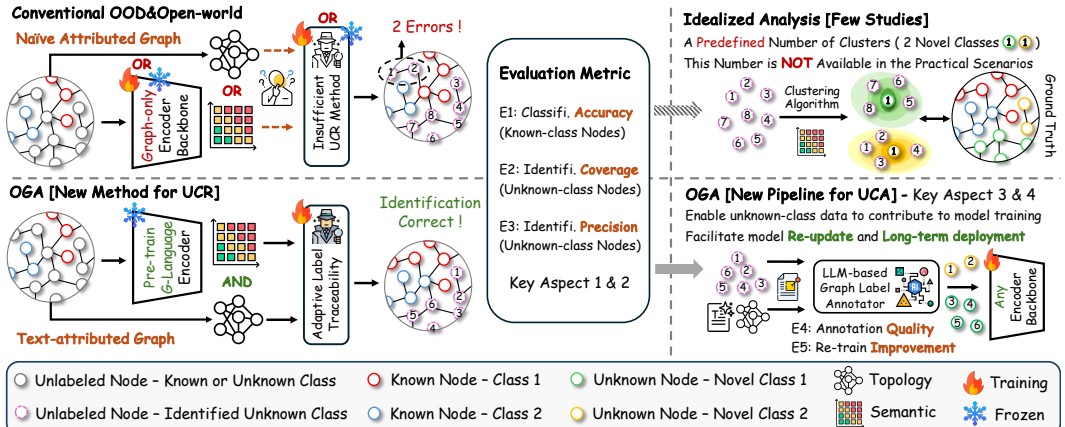

**Figure 1:** *A comparative overview of our proposed open-world learning pipeline and the conventional one. Our proposed OGA integrates LLM to introduce a new paradigm for unlabeled data uncertainty.*

focus on unlabeled data uncertainty. Therefore, we do not elaborate on them but instead provide a discussion in Appendix A.1.

Based on this, upon reviewing related OOD and open-world studies, we find that most of them are based on naive attributed graphs, where node and edge features are predefined by word embedding models. Despite their effectiveness, they face inherent limitations in the era of rapid advancements in LLMs and the increasing prevalence of TAGs, which are shown in Fig. 1 with two aspects:

❶ *Insufficient Method.* Prior studies typically employ a graph-only encoder (trainable or frozen) for embedding and then focus on either semantics or topology in isolation for unknown-class rejection. Specifically, semantic approaches aim to model feature discrepancies using Bayesian estimation or energy functions Zhao et al. (2020); Stadler et al. (2021); Gong & Sun (2024); Wu et al. (2023b); Yang et al. (2024); Um et al. (2025), and topology approaches seek to analyze structural variations through graph propagation or graph reconstruction Wu et al. (2020a); Song & Wang (2022b); Hoffmann et al. (2023b); Wang et al. (2024a); Ma et al. (2024); Wang et al. (2025). The limited information provided by naive attributed graphs and graph-only encoder hinders the compatible optimization.

☆ *Key Insights & Our Solutions.* ① In TAGs, the language is a bridge between semantics and topology, unlocking new opportunities for further breakthroughs. This means that rather than relying on an under-trained graph-only encoder, utilizing a well-pre-trained graph-language encoder can facilitate unbiased and enriched representations for all nodes Wen & Fang (2023); Kong et al. (2025); Yu et al. (2025). ② Based on these high-quality embeddings and informative TAGs, we propose adaptive label traceability (ALT), which achieves ontology representation learning by seamlessly integrating semantic-aware label boundaries and topology-aware regularization constraints in optimization. This approach effectively facilitates the classification of known-class nodes while enhancing the identification of unknown-class nodes.

❷ *Incomplete Pipeline.* Based on the above unknown-class rejection (UCR), prior pipelines primarily focus on two aspects: ① Enhancing the learning of known-class nodes Xu et al. (2023); Galke et al. (2021); Jin et al. (2024b); Wu et al. (2020a). ② Analyzing the unknown-class nodes under idealized clustering Wang et al. (2024c); Liu et al. (2023c); Hoffmann et al. (2023b). While these methods have achieved remarkable progress, their pipelines remain incomplete for practical applications. Specifically, the predefined number of unknown classes lacks utility and they fail to leverage these nodes effectively.

☆ *Key Insights & Our Solutions.* ① In TAGs, the abundant textual data and structural information present an opportunity for effective post-processing of unknown-class nodes, contributing to model re-updates and long-term deployment. This motivates us to propose graph label annotator (GLA), which leverages structural prompts to enable LLMs first to distill node description, and subsequently generate meaningful annotations. Meanwhile, we introduce a graph-oriented adaptive label fusion method, which improves annotation efficiency and quality (unknown-class annotation, UCA). ② By considering real-world deployment, our proposed open-world graph assistant (OGA) demonstrates strong practical utility. Based on this, to establish a comprehensive evaluation, we standardize 5 performance metrics from 4 key aspects, paving the way for future advancements in this field.

**Our contributions**. ❶ *New Perspective*. During our investigation, we found that existing studies focus on UCR. However, further leveraging unlabeled data (UCA) is also essential for model deployment and mitigating costly labor; unfortunately, this perspective has often been overlooked. ❷ *Innovative Approach*. Inspired by LLM and TAGs, we propose ALT, which effectively integrates semantic and topology optimization into a unified framework, enabling efficient known-class node classification and unknown-class node identification (UCR). Based on this, we introduce GLA, which utilizes structure-guided LLM to annotate unknown-class nodes for model re-updates (UCA). ❸ *New Pipeline*. We propose OGA, which contains ALT and GLA as the first LLM-enhanced open-world graph learning pipeline that integrates UCR with UCA, enhancing the practical utility of graph ML. It can be seamlessly integrated with any backbone to improve data efficiency. ❹ *SOTA Performance in 4 Key Evaluation Aspects*: ① Known-class Node Classification: OGA improves by 4.98% over the best baselines; ② Unknown-class Node Identification: OGA outperforms the best baselines by 6.2% in coverage and 4.6% in precision; ③ Annotation Efforts: OGA achieves comparable or superior semantic similarity to ground-truth annotations (Details in Sec. 4); ④ Post-annotation: OGA shows an average improvement of 10.1%, outperforming the unmarked graph and achieving 87%-105% of the ground-truth graph performance.

## 2 PRELIMINARIES

### 2.1 NOTATIONS AND PROBLEM FORMULATION

In this paper, we consider $\mathcal{G} = (\mathcal{V}, \mathcal{E})$ with $|\mathcal{V}| = n$ nodes, $|\mathcal{E}| = m$ edges. It can be described by an adjacency matrix $\mathbf{A} \in \mathbb{R}^{n \times n}$. As a TAG, $\mathcal{G}$ has node-oriented language descriptions, which are represented as T. Based on this, each node has a feature vector of size $f$ and a one-hot label of size $c$ generated by the language model and manual labeling, respectively. Formally, the feature and label matrix are $\mathbf{X} \in \mathbb{R}^{n \times f}$ and $\mathbf{Y} \in \mathbb{R}^{n \times c}$, and each node belong to one of $c$ classes. To simulate the unlabeled data uncertainty in open-world environments, we adopt a limited node labeling setting.

Specifically, a small subset of nodes $\mathcal{V}_l \subset \mathcal{V}$ is labeled, while the remaining nodes $\mathcal{V}_u = \mathcal{V} \setminus \mathcal{V}_l$ are unlabeled. For the labeled nodes, their labels are drawn from a subset $\mathcal{C}_k \subset \mathcal{C}$ of the complete, unseen label set, where $|\mathcal{C}| = c$. For each unlabeled node, it may belong to one of the following categories: ① known-class node: The potential label belongs to $\mathcal{C}_k$, which is already present in $\mathcal{V}_l$. ② unknown-class node: The potential label belongs to novel classes $\mathcal{C}_{uk} = \mathcal{C} \setminus \mathcal{C}_k$, which do not exist in $\mathcal{V}_l$. Based on this unlabeled data uncertainty, our goal is to develop a novel open-world pipeline that fully automates the processing of unlabeled data. The formal definition are as follows:

◆ **Unknown-Class Rejection (UCR).** To begin with, in the unlabeled set $\mathcal{V}_u$, we aim to identify the unknown-class nodes $\mathcal{V}_{uk}$ while performing classification for other known-class nodes $\mathcal{V}_k$. The evaluation metrics are as follows: ① Accuracy (Aspect 1): The proportion of known-class nodes that are correctly classified; ② Coverage (Aspect 2): The proportion of identified unknown-class nodes to the total number of unknown-class nodes. ③ Precision (Aspect 2): The proportion of nodes identified as the unknown class that genuinely belong to the unknown class.

◆ **Unknown-Class Annotation (UCA).** Then, we aim to annotate unknown-class nodes, enabling them to make substantive contributions to subsequent training. The evaluation metrics are as follows: ④ Quality (Aspect 3): The semantic similarity between each pair of classes in the current label set after annotation. Notably, the labels used in annotation are entirely generated by LLM and from the open domain. ⑤ Improvement (Aspect 4): The effectiveness of retraining the backbone on the unlabeled graph (lower bound), the annotated graph, and the ground-truth graph (upper bound).

### 2.2 UNKNOWN-CLASS REJECTION (UCR)

**Node-level Out-of-distribution Detection**. This task is crucial for anomaly detection and has gained increasing attention. Existing work mainly targets naive attributed graphs with graph-only encoders, providing limited self-supervised signals and weak node embeddings. Consequently, methods rely on either semantic- or topology-based optimization, struggling to integrate both. We categorize them into: ① semantic-oriented (Bayesian estimation, energy models) and ② topology-oriented (graph propagation, graph reconstruction).

Bayesian methods Zhao et al. (2020); Stadler et al. (2021) capture feature uncertainty but often fail when text-rich nodes and edges yield non-stationary distributions. Energy-based methods Wu et al. (2023b); Yang et al. (2024); Bao et al. (2024); Gong & Sun (2024); Chen et al. (2025); Um et al. (2025) score alignment with training distributions but depend heavily on graph-only encoders and suffer from high complexity and poor generalization. Topology-oriented approaches Song & Wang (2022b); Ma et al. (2024); Wang et al. (2024a; 2025) exploit propagation or reconstruction to analyze structural patterns, yet their limited use of semantics constrains performance on text-attributed graphs (TAGs).

**Conventional Open-world Graph Learning**. Methods typically focus on classifying known nodes and rejecting unknowns, often through post hoc designs with supervised graph-only encoders Galke et al. (2021); Wu et al. (2020a); Liu et al. (2023c); Hoffmann et al. (2023b). This neglects joint modeling of semantics and topology in TAGs, yielding biased unknown-node representations. Some extend by assuming predefined unknown classes Xu et al. (2023); Jin et al. (2024b); Wang et al. (2024c), but such idealized settings lack practicality. We argue that open-world UCR requires holistic solutions beyond these constraints.

### 2.3 Unknown-Class Annotation, UCA

As a core component of our proposed OGA, we conduct a comprehensive review of graph annotation. During our investigation, we noticed that only few related studies Chen et al. (2023) Zhang et al. (2024) explored how LLMs can be leveraged for node annotation. However, these approaches have the following limitations to varying degrees in open-world scenarios: ① They adopt a few-shot setting, where the ground-truth labels of unknown classes are provided to the LLM. This assumption restricts its practicality in the real world, where unknown classes are inherently undefined. ② They primarily focus on enhancing LLM annotation through chain-of-thought prompting, which neglects the relational information embedded in the graph structure.③ They lies in their reliance on a pre-defined number of clusters, which introduces strong assumptions about the label space and hinders their applicability in open-world or dynamically evolving graph scenarios. A comparison of our method and these approaches settings is provided in the Appendix A.2 Apart from the above study, some OOD detection methods Xu et al. (2025a) Xu et al. (2025b) leverage LLMs for semantic reasoning to identify potential OOD categories. However, their settings still diverge from the open-world graph learning scenario considered in this work. A detailed comparison is provided in Appendix A.3. This contrasts with our open-world setting, where we aim to handle unlabeled data uncertainty without prior knowledge.

### 3 Methods

#### 3.1 Overall of OGA

To advance the practical deployment of graph ML, we aim to tackle data uncertainty in open-world environments. This data-centric challenge is inherently difficult due to its intrinsic complexity and the lack of prior knowledge. However, the rapid advancement of LLMs and TAGs presents breakthrough opportunities. In this paper, we leverage a pre-trained graph-language encoder and abundant language descriptions in TAGs to incorporate prior knowledge, focusing on addressing the challenge of limited labeling. Specifically, we propose OGA, as illustrated in Fig. 2, the first framework to enable fully automated processing of unlabeled data in open-label domains. It overcomes the limitations of human-in-the-loop (i.e., real-time manual annotation), aligning with the practical demands of model deployment. In the following sections, we provide a detailed overview of the two key modules, ALT and GLA, as well as their utility within our proposed open-world pipeline, as introduced in Sec. 2.1.

#### 3.2 Adaptive Label Traceability for Unknown-Class Rejection

**Motivation.** Compared to naive attributed graphs, UCR in TAGs presents significant challenges. This is due to the fact that language descriptions introduce greater uncertainty in representation learning and the entanglement of semantics and topology hinders optimizations. To address this issue, our key insights are: ❶ *Pre-trained Graph-language Encoder*: We utilize its strong generalization capabilities to obtain unbiased, high-quality embeddings for all nodes in the graph, establishing a solid foundation for classification and identification. ❷ *Ontology Representation Learning*: For the first time, we introduce this concept into UCR, leveraging the flexible concepts, entities, and relations in ontology to enable adaptive open-label domain representation learning as follows:

**Figure 2:** *The overview of our proposed OGA.*

◆ *What is Ontology Represented in UCR?* Ontology is a formal and explicit representation of concepts, entities, and their interrelations within a domain. In the context of UCR: ① The domain refers to the open-label setting; ② Concepts denote label classes, analogous to class-wise prototypes (via class-specific embedding averaging); ③ Entities are nodes in the graph; ④ Interrelations capture latent dependencies between concepts and entities. We provided theoretical analysis in theorem 1, 2.

◆ *Why is Ontology Essential in UCR?* Unlike prior methods that statically generate prototypes from labeled data, open-label uncertainty requires dynamic concept construction. Unlabeled nodes potentially belonging to known classes must contribute to concept formation. However, due to prediction uncertainty, soft labels alone are unreliable. Since concept-entity relations are implicit in open-world settings, we formulate an optimization objective to adaptively uncover these relations. This enables robust class boundary formation in the ontology space, supporting both known-class classification and unknown-class rejection.

◆ *How is Ontology Representation Learning Achieved in UCR?*

❶ *Entity-Entity*: Firstly, we aim to reveal entity-entity dependency from a topology perspective, thereby exhibiting an affinity toward specific known classes for concept generation. To achieve this without distorting the embedding space, we utilize the label supervision and personalized PageRank:

$$\mathbf{E}=\text{LLM-Graph}\left(\mathcal{G}, \mathbf{T}\right), \ \tilde{\mathbf{E}}=(\mathbf{I}+\kappa\mathbf{P})\cdot\mathbf{E}=\sum_{l=0}^{k}\sum_{i=0}^{l}\left(\mathbf{E}^{(0)}+\kappa\cdot w_i\cdot\left(\hat{\mathbf{D}}^{r-1}\hat{\mathbf{A}}\hat{\mathbf{D}}^{-r}\right)^i\mathbf{E}^{(l)}\right), \quad (1)$$

where $\kappa$ is the fine-tuned intensity factor for entity embedding $\mathbf{E}$, $\mathbf{P}$ is the trainable $k$-step graph propagation equation, serving as the paradigm to model node proximity measures. For a given node $u$, a node proximity query yields $\mathbf{P}(v)$, representing the relevance of $v$ with respect to $u$. The learnable weight sequence $w_i$ and kernel coefficient $r$ affect transport probabilities for pair-wise nodes.

❷ *Entity-Concept*: Subsequently, we aim to dynamically and adaptively generate concepts $\mathbf{C} \subset \mathcal{C}_k$ by labeled data and structural information. The dynamicity stems from the iterative selection of entity sets $\mathbb{N}$, while the adaptivity arises from trainable weights $w$. The above process is defined as:

$$\mathbf{C}_c = \frac{1}{|\mathcal{S}_c|}\sum_{\tilde{e}_i\in\mathcal{S}_c}\sum_{j\in\mathbb{N}(i)}\frac{(\mathbf{I}+w_j)\,\tilde{e}_j}{|\mathbb{N}(i)|}, w_j=\frac{\exp\left[\delta\left(\mathbf{H}_j\right)\right]}{\sum_{l=1}^{|\mathbb{N}(i)|}\exp\left[\delta\left(\mathbf{H}_j^{(l)}\right)\right]}, \mathbf{H}_j=\text{MLP}\left(\tilde{e}_1||\cdots||\tilde{e}_{|\mathbb{N}(i)|}\right), \quad (2)$$

where $\mathcal{S}_c$ is the set of data sets annotated with class $c$ and $\mathbb{N}(i)$ is the neighborhood sets (i.e., all nodes within $K$-hop proximity) of node $i$. In our implementation, we set $k = [1...5]$ and randomly sample $k$-hop neighbors in each training epoch. This strategy ensures that the concept generation propagates across the entire graph, rather than being restricted to the limited labeled data, expressing diversity. Based on this, trainable $w_j$ is employed for adaptive entity aggregation, expressing generalizability.

❸ *Concept-Entity*: At this stage, we have established class boundaries (i.e., concept representation in the ontology space). Based on this, we employ a distance function (e.g., Euclidean distance) to

quantify the relevance between each entity and all concepts. Then, we apply $\lambda$-sharpness softmax to this measurement to classify known-class nodes. As for unknown-class node identification, we introduce a confidence threshold $\epsilon$ based on the intuition that such nodes often exhibit a smooth class probability distribution (e.g., entropy measurement). The above process is defined as:

$$\hat{y}_i = \begin{cases} \arg\max_c \mathcal{D}_{i,c}, & \text{if } \mathrm{Conf}\,(\mathcal{D}_{i,c}) \geq \epsilon \\ \text{Unknown-class}, & \text{otherwise.} \end{cases}, \mathcal{D}_{i,c} = \frac{\exp(-\lambda\|\tilde{e}_i - \mathbf{C}_c\|_2)}{\sum_{c\in\mathcal{C}_k}\exp(-\lambda\|\tilde{e}_i - \mathbf{C}_c\|_2)}. \tag{3}$$

❹ *Optimizations*: To integrate the above modules into an end-to-end framework, we design an optimization objective that jointly incorporates semantics and topology in the ontology space. Specifically, we combine a semantic cross-entropy loss, a topology-aware smoothness loss, and a margin-based separation loss with theorem 3:

$$\mathcal{L} = \mathcal{L}_{ce} + \alpha\mathcal{L}_{smooth} + \beta\mathcal{L}_{separate},$$
$$\mathcal{L}_{ce} = \sum_{i\in\mathcal{V}_l}\sum_j \mathbf{Y}_{ij}\log\left(\mathrm{softmax}(\mathcal{D}_{ij})\right),$$
$$\mathcal{L}_{smooth} = \mathrm{trace}(\mathcal{D}^T(\mathbf{I}-\mathbf{P})\mathcal{D}) + \|\mathcal{D}-\mathcal{Y}\|_F^2, \tag{4}$$
$$\mathcal{L}_{separate} = -\left[\sum_{i\neq j}\frac{\mathbf{C}_i\cdot\mathbf{C}_j}{\|\mathbf{C}_i\|\|\mathbf{C}_j\|} + \min\left(\max_{i\neq j}\mathcal{M}(\mathbf{C}_i,\mathbf{C}_j),\theta\right)\right],$$

where $\mathcal{M}$ is a distance-based margin function and $\theta$ prevents unbounded margin growth. The separation loss encourages inter-class distinction in the concept space. The smoothness loss refines prediction residuals by promoting local consistency via graph proximity and alignment with labeled data. For a detailed comparison with prior UCR objectives, see Appendix A.4.

### 3.3 Graph Label Annotator for Unknown-Class Annotation

**Motivation.** After UCR, we are committed to achieving structure-guided LLM annotation. The key insights are: ❶ *Topology-aware Semantic In-context*: Inspired by graph mining and the divide-and-conquer paradigm, we integrate text semantic similarity into off-the-shelf community detection to obtain valuable in-context for annotation efficiency. ❷ *Multi-granularity Community Annotation*: To enhance annotation quality, we employ a two-stage LLM inference based on the structural metrics: intra-community coarse-grained distillation and inter-community fine-grained fusion.

**Topology-aware Semantic In-context.** To achieve this, we aim to develop a community detection algorithm tailored for TAGs. The core idea is to extend conventional methods by additionally integrating textual semantics into a unified and weight-free optimization objective $\mathcal{Q}$. In our implementation, we adopt a modularity-based objective enhanced with semantic similarities, formulated as follows:

$$\mathcal{Q} = \frac{1}{2m}\sum_{i,j}\left[\mathbf{A}_{ij} + \gamma\cdot\frac{\tilde{e}_i^T\tilde{e}_j}{\|\tilde{e}_i\|\cdot\|\tilde{e}_j\|} - (1-\gamma)\cdot\frac{d_id_j}{2m}\right]\delta(c_i,c_j), \tag{5}$$

where $d_i$ represents the degree of node $v_i$, $\delta(c_i,c_j)$ equals 1 if $v_i$ and $v_j$ belong to the same community, and 0 otherwise. The $\gamma$ balances the contributions of semantic measurements. After that, within each community, we compute the degree of each node as additional in-context and classify them as low-degree or high-degree based on the median. This is used for degree-informed efficient LLM inference strategy. Specifically, we prioritize low-degree nodes for annotation, while high-degree nodes leverage annotated neighbors, enabling annotation without invoking the LLM. The core intuition follows the homophily assumption: low-degree nodes, lacking neighborhood prior knowledge, need LLM for annotation, whereas high-degree nodes can benefit from neighbors. Notably, since low-degree nodes are sparse, this approach significantly reduces LLM inference costs. Theoretical analysis in 4.

**Multi-granularity Community Annotation.** To this end, we define the scope (community) and order (degree) of LLM annotation. Then, guided by structural metrics, we sequentially implement annotation distillation within each community and annotation fusion across communities. To begin with, the degree-guided annotation generation for each node within community $\mathcal{P}$ is formulated as:

$$\tilde{y}_i = \begin{cases} \text{LLM-Annotation}\,(\mathrm{T}_i, \{\mathrm{T}_j \mid v_j \in \mathcal{N}(v_i)\}), & \text{if } d_i < \bar{d} \\ \text{Allocation}\,(\mathrm{T}_i, \{\mathrm{T}_j, \tilde{y}_j \mid v_j \in \mathcal{N}(v_i)\}), & \text{if } d_i \geq \bar{d}. \end{cases}, \tag{6}$$

At this stage, we incorporate limited neighboring text to enhance low-degree node annotations without significant overhead. Based on this, we identify the top-$\Phi$ representative nodes within each community via topology and semantic measurement and perform annotation distillation as follows:

$$v_\phi = \arg \max_{\phi \in \mathcal{P}} \frac{1}{|\mathcal{P}|} \sum_{i \neq j} \left[ \frac{|\mathcal{N}(v_i) \cap \mathcal{N}(v_j)|}{|\mathcal{N}(v_i) \cup \mathcal{N}(v_j)|} + \mathcal{M}(\tilde{e}_i, \tilde{e}_j) \right], \tilde{y}_\mathcal{P} = \text{LLM-Distill}\left(\tilde{y}_1, ..., \tilde{y}_\Phi\right). \quad (7)$$

This annotation $\tilde{y}_\mathcal{P}$ is shared by all nodes within the community $\mathcal{P}$. After that, we iteratively fusion the most similar community-level annotations to reduce redundancy until the number of annotations decreases to a predefined threshold, which in our implementation is determined by the number of known classes. The above annotation fusion and similarity measurement can be formally defined as:

$$\text{Sim}\left(\mathcal{P}_i, \mathcal{P}_j\right) = \frac{1}{|\mathcal{P}_i||\mathcal{P}_j|} \sum_{m \in \mathcal{P}_i} \sum_{n \in \mathcal{P}_j} \mathcal{M}(\tilde{e}_m, \tilde{e}_n), \ \tilde{y}^\star_{\mathcal{P}_{ij}} = \text{LLM-Fusion}\left(\tilde{y}_{\mathcal{P}_i}, \tilde{y}_{\mathcal{P}_j}\right). \quad (8)$$

To this end, we generate a limited set of $\tilde{y}^\star_\mathcal{P}$ for final annotation. As an example, for a fusion label $\tilde{y}^\star_{\mathcal{P}_{ijk}} = \text{LLM-Fusion}(\tilde{y}^\star_{\mathcal{P}_{ij}}, \tilde{y}_{\mathcal{P}_k})$, where $i, j, k$ are community IDs, the $\tilde{y}^\star_{\mathcal{P}_{ijk}}$ is allocated to all nodes within these communities, thereby achieving annotation. For the detailed implementation of the structure-guided prompt templates for LLM inference in Eq. (6-8), please refer to Appendix A.5.

### 3.4 THEORETICAL ANALYSIS

We provide theoretical guarantees for the design of ALT and GLA by analyzing concept construction and community optimization.

**Theorem 1.** *Let embeddings $\tilde{E} = \{\tilde{e}_i\}$ be generated by a Lipschitz-continuous encoder over a compact manifold $\mathcal{M}$, and let class concepts $\{C_c\}$ be aggregated via a propagation matrix $P$ with Dirichlet energy bounded by $\delta$. If intra-class variance is bounded by $\sigma^2$, then: $\text{rank}(\text{span}(\{C_c\})) \leq r$, $\mathbb{E}_{i \in \mathcal{S}_c}\|\tilde{e}_i - C_c\|^2 \leq \sigma^2 + \varepsilon(\delta)$, and $\|C_i - C_j\|^2 \geq \Delta$ with $\cos \angle(C_i, C_j) \leq \rho < 1$.*

This shows that ALT constructs a concept space that is low-rank, compact, and discriminative—suitable for reliable classification and rejection (Proof in Appendix A.6.1).

**Theorem 2.** *Let class-wise concept vectors $\{C_c\}$ be constructed via structure-only propagation in ALT, forming implicit hyperspheres with bounded Dirichlet energy $\delta$. Then the total concept volume satisfies a compression bound, and the inter-class redundancy is exponentially suppressed when $\|C_i - C_j\| \geq \theta_{\min}$.*

This ensures ALT achieves compact and separable class representations purely from topology, enabling robust rejection and generalization without semantic supervision (Proof in Appendix A.6.2).

**Theorem 3.** *Let node $i$ be classified by a $\lambda$-sharpened softmax over concepts $\{C_c\}$. If rejected under confidence threshold $\epsilon$, then its entropy satisfies $H(D_i) \geq \log|C_k| - \frac{1}{\lambda} \log \frac{1-\epsilon}{\epsilon}$.*

This quantifies ALT's rejection uncertainty and supports controllable unknown-class identification (Proof in Appendix A.6.3).

**Theorem 4.** *Let communities $\{P_k\}$ be obtained by optimizing semantic-enhanced modularity $Q$ with parameter $\gamma$. Then intra- and inter-community similarities satisfy $S_{intra} \geq \eta(\gamma, \zeta)\bar{s}$ and $S_{inter} \leq (1 - \eta(\gamma, \zeta))\bar{s}$, where $\bar{s}$ is the global average similarity and $\zeta$ the structural-semantic coherence. Moreover, $\lim_{\gamma \to 1} \eta(\gamma, \zeta) = 1$.*

This ensures GLA promotes semantic cohesion within and separation across communities, enabling more efficient and accurate annotation (Proof in Appendix A.6.4).

## 4 EXPERIMENT

In this section, we conduct a wide range of experiments and aim to answer: **Q1: Effectiveness.** Compared with other SOTA baselines, can OGA achieve superior UCR and UCA performance? **Q2: Interpretability.** If OGA is effective, what factors contribute to the success of ALT and GLA? **Q3: Robustness.** How sensitive is OGA to hyperparameters, and how does it perform in complex data scenarios? **Q4: Efficiency.** What is the running efficiency of OGA? Due to space constraints, please refer to Appendix A.7 (exp environments), Appendix A.8 (datasets), Appendix A.9 (baselines), Appendix A.10 (evaluation protocols), and Appendix A.11 (hyperparameters) for more details.

**Table 1:** *UCR performance in Aspect 1.* **Red**: *1st,* **Blue**: *2nd,* **Orange**: *3rd (%).*

| Dataset | Cora | arXiv | Children | Wikics |
|---|---|---|---|---|
| ORAL | 87.54±0.62 | 74.62±0.58 | 37.13±0.53 | 79.83±0.20 |
| OpenWgl | 87.01±0.94 | 62.85±1.03 | 30.72±0.89 | 82.36±0.28 |
| OpenIMA | 78.52±0.17 | OOM | OOM | 86.12±0.45 |
| OpenNCD | 80.77±0.31 | OOM | OOM | 54.52±0.84 |
| IsoMax | 77.76±0.08 | 76.38±0.07 | 38.72±0.09 | 82.26±0.56 |
| GOOD | 87.62±0.28 | 76.03±0.70 | 50.44±0.71 | 57.97±0.17 |
| gDoc | 30.37±0.27 | 75.31±0.49 | 47.55±0.40 | 74.74±0.36 |
| GNN_safe | 85.85±0.25 | 72.20±0.91 | 50.66±0.51 | 82.55±0.61 |
| OODGAT | 80.00±0.97 | 55.80±0.87 | 39.44±0.95 | 78.39±0.38 |
| ARC | 71.02±0.20 | 60.22±0.23 | 37.54±0.72 | 72.13±0.74 |
| EDBD | 76.85±0.48 | 65.34±0.16 | 36.94±0.40 | 70.34±0.53 |
| **Ours(OGA)** | **90.02±0.24** | **78.39±0.09** | **51.10±0.11** | **87.53±0.18** |

**Table 2:** *UCR performance in Aspect 2. Each item is expressed as* **Coverage/Precision***(%).*

| Dataset | Cora | arXiv | Children | Wikics |
|---|---|---|---|---|
| ORAL | 85.2 / 33.6 | 73.3 / 31.4 | 54.0 / 12.7 | 32.8 / 7.2 |
| OpenWgl | 37.9 / **91.3** | 64.1 / **72.4** | 37.8 / 12.3 | 59.4 / 18.6 |
| OpenIMA | 80.5 / 64.0 | 80.6 / **64.3** | OOM | 69.1 / **33.9** |
| OpenNCD | 27.7 / **87.1** | OOM | OOM | 48.5 / 19.4 |
| IsoMax | 67.3 / 65.7 | 59.2 / 52.4 | 50.1 / 13.0 | 38.6 / 13.5 |
| GOOD | 72.8 / 71.5 | **88.0** / 51.8 | 81.4 / 12.9 | 78.3 / 17.3 |
| gDoc | 71.9 / 71.2 | 80.6 / **64.3** | 75.2 / 9.3 | 79.2 / **59.6** |
| GNN_Safe | **90.0** / 43.1 | 85.4 / 40.0 | **88.6** / **14.0** | **87.7** / 30.2 |
| OODGAT | **89.5** / 51.6 | **96.1** / 22.7 | **86.7** / **21.2** | **83.4** / 17.8 |
| ARC | 83.4 / 71.5 | 65.3 / 33.6 | 65.7 / 11.8 | 43.1 / 11.7 |
| EDBD | 54.3 / 72.0 | 62.2 / 22.6 | 39.5 / 16.8 | 39.7 / 16.5 |
| **Ours (OGA)** | **94.2 / 82.3** | **91.7 / 60.5** | **96.9 / 24.3** | **96.4 / 48.9** |

## 4.1 PERFORMANCE COMPARISON (Q1)

**UCR Performance.** Table 1-2 consistently show that OGA achieves the best performance in most cases across the two aspects outlined in Sec. 2.1. Specifically, in Aspect 1 (Accuracy), Table 1 demonstrates that OGA achieves superior performance, particularly on the Cora and WikiCS datasets. In Aspect 2 (Coverage/Precision), Table 2 highlights the difficulty of achieving a balanced trade-off, as existing methods tend to perform well in only one metric. However, OGA leverages ontology representation learning to adapt to an open-label domain, thereby achieving satisfactory balanced performance effectively. Please refer to Appendix A.12.1 and A.12.2 for more experimental results.

**UCA Performance.** We first clarify the notation used in the benchmark for Aspect 3 (Quality): "k" denotes known-class labels, "u" represents ideal labels for unknown classes (for evaluation only), and "g" denotes labels generated by OGA. As shown in Table 3, GLA achieves clear semantic separation in open-label domains by leveraging topology-guided multi-granularity annotation. More results are provided in Appendix A.12.3. For Aspect 4 (Efficiency), Table 4 shows that OGA significantly restores GNN performance under limited labeling (e.g., GCN on PubMed improves from 47.12% to 87.15%, approaching the upper bound of 87.49%). Additional results are in Appendix A.12.4.

**Table 3:** *UCA performance in Aspect 3.*

| Similarity | Cora | arXiv | Children | Wikics |
|---|---|---|---|---|
| **k to k (baseline)** | 75.01 | 80.02 | 71.04 | 73.09 |
| **k to g (↓)** | 35.12 | 37.34 | 32.10 | 33.17 |
| **g to u (↑)** | 60.14 | 63.09 | 57.12 | 58.06 |

**Table 4:** *UCA performance in Aspect 4.*

| GCN | Cora | Citeseer | Pubmed |
|---|---|---|---|
| Lower | 61.62 | 58.67 | 47.12 |
| Ours | 78.15 | 65.72 | 87.15 |
| Upper | 85.56 | 72.71 | 87.49 |

| GAT | Cora | Citeseer | Pubmed |
|---|---|---|---|
| Lower | 60.52 | 58.51 | 46.33 |
| Ours | 79.26 | 69.50 | 86.45 |
| Upper | 87.04 | 74.08 | 86.19 |

## 4.2 ABLATION STUDY AND IN-DEPTH INVESTIGATIONS (Q2)

**ALT Part (UCR Performance).** As shown in Table 5, both Concept Modeling (CM) and Topology-aware Propagation (TP) play essential roles in the performance of OGA. Specifically, CM defines and refines class boundaries within the ontology space, enhancing the model's ability to accurately distinguish between known and unknown classes. TP incorporates graph topology into the representation learning process, enabling node embeddings to capture both semantic meaning and structural context. Additional details regarding the ALT ablation study and the effectiveness of the entropy-based softmax approach are provided in Appendix A.13.1 and Appendix A.13.2.

**GLA Part (UCA Performance).** To evaluate the effectiveness of topology-guided community detection in supporting multi-granularity semantic mining for graph annotation, we conduct an in-depth analysis from three perspectives—redundancy (Re), consistency (Con), and accuracy (Acc)—as presented in Table 6. The results demonstrate that high-quality topological context enables seamless and efficient integration between graph annotation and LLM inference. The definitions and computation of these three metrics, along with additional experimental results on the GLA ablation study and a case study, are provided in Appendix A.13.3 and A.13.4.

**Table 5:** *Ablation results of ALT.*

| Method | Cora | Citeseer | Pubmed |
|---|---|---|---|
| w/o CM | 41.25 / 62.31 | 39.24 / 19.12 | 52.10 / 21.75 |
| w/o TP | 73.15 / 73.52 | 70.22 / 52.81 | 69.83 / 61.27 |
| **Full ALT** | **94.87 / 82.66** | **93.00 / 65.19** | **79.76 / 98.53** |

**Table 6:** *Ablation results of GLA.*

| Dataset | Full GLA | | | w/o Semantic Community | | |
|---|---|---|---|---|---|---|
| | Re | Con | Acc | Re | Con | Acc |
| Citeseer | 9 | 0.84 | 72.1% | 17 | 0.70 | 67.3% |
| Pubmed | 5 | 0.86 | 86.0% | 11 | 0.73 | 81.2% |

### 4.3 ROBUSTNESS ANALYSIS OF HYPERPARAMETERS AND DATA DISTRIBUTIONS (Q3)

**ALT Part (Model Perspective).** In this section, we investigate the impact of the $\lambda$ in Eq. 3, which directly influences the sharpness of the decision boundary and thus plays a critical role in both UCR and UCA. As shown in Figure 3, increasing $\lambda$ enhances model robustness by improving class separability and rejection capability. However, excessively large values of $\lambda$ lead to overconfidence. A moderate range of $\lambda$ provides a more favorable trade-off between UCR and UCA. Although the ALT module involves multiple hyperparameters, due to space constraints, we focus on the most influential one, $\lambda$, in the main text. More details about $\lambda$, impact of unknown-class on ALT and sensitivity analyses for $\alpha$ and $\beta$ in Eq. 4 are presented in Appendix A.14.1, A.14.2 and A.14.3.

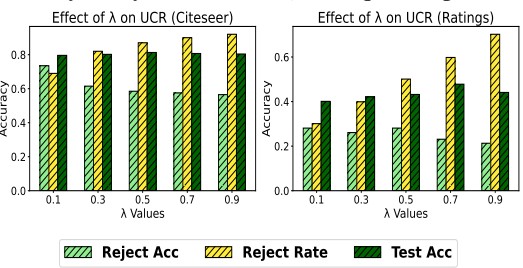
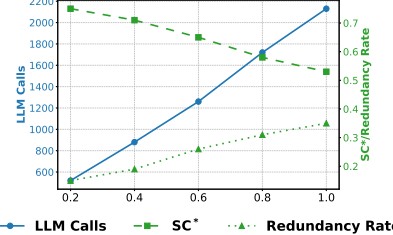

**Figure 3:** *ALT sensitivity analysis (model).*

**Figure 4:** *GLA sensitivity analysis (data).*

**GLA Part (Data Perspective).** In this section, we analyze the impact of the proportion of unlabeled known-class nodes on annotation, as illustrated in Figure 4 . The semantic consistency metric $SC^*$—defined as the average pairwise semantic similarity among generated labels with known ground-truth classes—exhibits a steady decline. The Redundancy Rate measures the proportion of semantically overlapping labels among all community-level annotations; a higher redundancy rate suggests an increased presence of duplicate or semantically similar labels across communities, thereby indicating reduced annotation effectiveness. In addition to data considerations from the deployment, the impact of $\gamma$ in Eq. 5 is analyzed in Appendix A.14.4.

### 4.4 EFFICIENCY COMPARISON (Q4)

**Table 7:** *Running time (in seconds).*

| Dataset | Cora | arXiv | Children | WikiCS |
|---|---|---|---|---|
| OpenWGL | 34.93 | 2223.27 | 727.52 | 1235.24 |
| IsoMax | 4.21 | 38.12 | 25.33 | 12.79 |
| GOOD | 13.12 | 144.66 | 127.64 | 53.17 |
| Ours | **8.91** | **99.23** | **73.83** | **29.48** |

**Table 8:** *LLM call comparison.*

| Dataset | Cora | arXiv | Children | WikiCS |
|---|---|---|---|---|
| Pure Calls | 2,708 | 169,343 | 76,875 | 11,701 |
| **GLA Calls** | **329** | **17,354** | **7,973** | **1,285** |
| (with) Com Count | 52 | 411 | 277 | 108 |
| **Reduction** | **87.8% ↓** | **89.8% ↓** | **89.6% ↓** | **89.0% ↓** |

**ALT Part (UCR Running Time).** As shown in Table 7, the ALT module substantially reduces runtime compared to OpenWGL, particularly on large-scale arXiv, due to its lightweight graph propagation. While IsoMax incurs lower computational overhead, its overly simplistic design leads to significantly inferior performance. In contrast, OGA achieves both high efficiency and superior effectiveness. More theoretical complexity analysis of ALT is provided in Appendix A.15.1.

**GLA Part (UCA LLM Calls).** Table 8 shows that GLA reduces LLM calls by over 87% across all datasets through a topology-prompted strategy. Rather than performing node-wise inference, GLA conducts batch annotation over a limited number of communities ("Com Count"), substantially decreasing LLM usage while maintaining validity. Notably, this efficiency gain does not come at the expense of performance. More details on the efficiency of GLA are provided in Appendix A.15.2.

### 5 CONCLUSION

In this paper, we propose OGA, the first fully automated data annotation framework based on LLMs to address unlabeled data uncertainty in open-world graph learning. By integrating both semantic and topology-based methods, OGA effectively handles unknown-class nodes, opening up a new research direction and promoting the deployment of graph machine learning in real-world scenarios. Our approach outperforms SOTA methods in both classification accuracy and rejection performance, demonstrating its significant contributions to the field.In the future, we will pay more attention to the scalability of OGA on large-scale graphs or special legends such as temporal or spatial graphs.

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

# A   APPENDIX

**Outline**

The appendix is organized as follows:

**A.1** Unlabeled data uncertainty and streaming data management

**A.2** Comparison between Label-free LLM Annotations and OGA in Open-world

**A.3** Comparison between OGA and Recent Graph OOD Detection Methods

**A.4** Ontology representation learning and prior approaches for UCR

**A.5** Structure-guided prompt template for LLM annotation

**A.6** Theoretical Analysis

**A.7** Experimental environment

**A.8** Dataset

**A.9** Baseline

**A.10** Evaluation

**A.11** Hyperparameter Settings

**A.12** Experimental Performance

**A.13** Interpretability Analysis

**A.14** Robustness Analysis

**A.15** Efficiency Analysis

**A.16** Declaration of Writing Assistance Technologies

## A.1   UNLABELED DATA UNCERTAINTY AND STREAMING DATA MANAGEMENT

In recent years, graph ML has witnessed rapid advancements, particularly in ideal experimental settings characterized by high-quality feature engineering and abundant human annotations. Specifically, numerous model frameworks have demonstrated remarkable predictive performance on benchmark evaluations. While these models have proven to be effective in controlled environments, they often struggle when deployed in real-world applications, where complex and dynamic data conditions prevail. We collectively refer to these challenges as data uncertainty in open-world scenarios.

The issue of data uncertainty is inherently complex, as it evolves with deployment environments, making it difficult to establish a precise and universally accepted definition. However, rather than attempting to formalize a rigid definition, we can decompose this broad concept into more tangible subproblems that enable targeted discussions. For instance, we focus on two critical subproblems: the challenge of large-scale unlabeled data and the management of streaming data, both of which represent significant facets of data uncertainty in real-world settings.

Upon reviewing existing studies in the open-world environments, we observe that the challenge of unlabeled data uncertainty is closely related to node-level out-of-distribution (OOD) and conventional

open-world graph learning Wang et al. (2024a; 2025); Bao et al. (2024); Stadler et al. (2021); Um et al. (2025); Wu et al. (2023b); Ma et al. (2024); Chen et al. (2025); Song & Wang (2022b); Gong & Sun (2024); Yang et al. (2024); Zhao et al. (2020); Xu et al. (2023); Liu et al. (2023c); Galke et al. (2021); Jin et al. (2024b); Wang et al. (2024c); Wu et al. (2020a); Hoffmann et al. (2023b). Meanwhile, streaming data management exhibits strong connections with continual, incremental, and lifelong graph learning Choi et al. (2024); Qi et al. (2025); Kou et al. (2020); Lin et al. (2023); Niu et al. (2024); Pang et al. (2024); Tang & Matteson (2020); Wang et al. (2020); Zhou & Cao (2021); Hoang et al. (2023); Zhang et al. (2022); Daruna et al. (2021); Rakaraddi et al. (2022). Although they all aim to enhance the practical applicability of graph ML by equipping them with the ability to handle complex data environments, they approach data uncertainty from fundamentally different perspectives.

Specifically, unlabeled data uncertainty assumes a static data environment, where the complete dataset has already been collected. The primary challenge in this setting arises from the lack of annotations due to costly labor. Additionally, the label space for unlabeled data may originate from an open-world setting, meaning that unknown label classes could exist beyond the initially defined label classes. In contrast, streaming data management assumes a dynamic data environment, where new data continuously arrives over time. At any given time step, prior data is either inaccessible or only available with limited access, due to storage constraints and privacy considerations. Furthermore, within each time step, the arriving dynamic data is typically associated with a specific task, and in most studies, the task ID is assumed to be known during training. While some works adopt a few-shot learning setting, they generally assume that for novel label classes, a small number of labeled examples are provided to facilitate learning. Therefore, rather than treating them as a monolithic problem, it is crucial to analyze them within their respective contexts.

In a nutshell, the effective deployment of graph ML in real-world applications necessitates addressing a wide range of data uncertainty challenges, including unlabeled data uncertainty and streaming data management, among others. Encouragingly, the graph ML community has recently begun to recognize the significance of these issues, with an increasing number of researchers actively exploring potential solutions. While it is imperative to examine different aspects of data uncertainty separately at the current stage, our ultimate goal should be the development of human-out-of-the-loop graph ML systems that exhibit strong robustness in the face of complex and evolving data environments.

## A.2 COMPARISON BETWEEN LABEL-FREE LLM ANNOTATIONS AND OGA IN OPEN-WORLD

① **Fixed-class assumption.** Existing label-free LLM annotation methods, such as LLM-GNN Chen et al. (2023), typically assume the availability of a fixed label space. They either perform clustering over node embeddings with a pre-defined number of clusters or adopt few-shot prompting with manually selected examples from known classes. This design implicitly assumes prior knowledge of the label space, which fundamentally contradicts the open-world setting where unknown classes are inherently undefined, noisy, or dynamically emerging. Consequently, these methods struggle to generalize in practical scenarios such as long-tail discovery or zero-shot classification on evolving graphs.

② **Neglect of structural semantics.** Another key limitation of these methods lies in their limited use of graph topology. Most existing approaches rely on node-wise textual prompts or local confidence-based filtering, yet they neglect the global structural patterns and semantic correlations among nodes. This often leads to isolated and redundant LLM queries, which fail to exploit the relational inductive bias inherent in graph-structured data. As a result, their annotation process suffers from low efficiency, reduced consistency, and poor scalability to large or sparsely labeled graphs.

③ **Dependency on predefined clustering.** A representative method like Cella Zhang et al. (2024) heavily depends on subspace clustering to select representative nodes for LLM annotation. The number of clusters $K$ must be specified in advance, and is typically set to match the number of ground-truth classes or a slightly larger value. This practice assumes strong prior knowledge of the label distribution, which is rarely available in open-world or continuously evolving graph environments. In contrast, our proposed **OGA** framework removes this dependency by leveraging dynamic ontology representation learning. It constructs semantic concepts directly from unlabeled data through structure-aware propagation and entity-entity aggregation, enabling autonomous discovery of latent class

boundaries without requiring pre-defined taxonomy. This design grants OGA strong adaptability to unknown or emerging classes, making it better aligned with real-world deployment scenarios.

## A.3 COMPARISON BETWEEN OGA AND RECENT GRAPH OOD DETECTION METHODS

① **Incomplete modeling of unknown classes.** Recent graph OOD detection methods, such as LEGO-Learn Xu et al. (2024) and GOE-LLM Xu et al. (2025a), primarily focus on *unknown-class rejection* (UCR), aiming to distinguish in-distribution (ID) nodes from out-of-distribution (OOD) nodes. While this aligns with the first step of open-world learning, these methods do not further utilize the rejected nodes for model enhancement. Specifically, they lack the ability to annotate, integrate, and retrain on previously unknown-class nodes, thus failing to form a complete learning loop. In contrast, our proposed **OGA** introduces the *unknown-class annotation* (UCA) module, which integrates LLM-guided community annotation into the pipeline, enabling dynamic concept discovery and long-term model improvement — a critical requirement for real-world deployment.

② **Reliance on real or pseudo OOD supervision.** Methods like GOE-LLM Xu et al. (2025a) and GLIP-OOD Xu et al. (2025b) assume access to either pseudo-OOD samples generated via LLMs or LLM-generated pseudo labels for synthetic exposure. However, both approaches implicitly require external intervention: either the manual specification of ID labels or LLM feedback on candidate OOD nodes. Such assumptions are inconsistent with the nature of unlabeled data in open-world graphs, where neither ID nor OOD supervision can be guaranteed. In contrast, **OGA** autonomously distinguishes and annotates unknown-class nodes without assuming any predefined OOD samples or auxiliary labels. By leveraging ontology representation learning over structure-text fused embeddings, OGA supports fully automated processing under realistic and minimal assumptions.

③ **Lack of semantic-topological integration.** A common drawback across existing methods lies in their underutilization of graph topology. LEGO-Learn Xu et al. (2024) and GOE-LLM Xu et al. (2025a) rely on node-wise GNN scoring or local ID/OOD classifiers, while GLIP-OOD Xu et al. (2025b) uses LLMs to generate label names without exploiting the underlying structure. These designs miss the opportunity to integrate *semantic signals* from text attributes with *topological regularities* from graph edges. In contrast, **OGA** explicitly couples semantic and topological factors through adaptive label traceability (ALT), which aligns concept formation with personalized graph propagation, modularity-augmented community detection, and confidence-aware rejection. This synergy significantly enhances label consistency and annotation coverage, particularly in sparse or dynamically evolving graphs.

④ **Static design and limited adaptability.** Many baseline frameworks assume static class sets (e.g., LEGO-Learn assumes fixed $C$-way classifiers with class-balanced selection), which impedes adaptability to new, unseen classes. GLIP-OOD attempts zero-shot extension but still depends on synthetic OOD label generation, which may fail to cover latent class granularity. In contrast, **OGA** embraces the *dynamic ontology* paradigm: concepts (i.e., class prototypes) are constructed directly from evolving data, without requiring static label names or cluster numbers. This dynamic design allows OGA to handle emerging classes and long-tail distributions effectively, facilitating seamless adaptation in open-world graph environments.

## A.4 ONTOLOGY REPRESENTATION LEARNING AND PRIOR APPROACHES FOR UCR

The UCR problem in the context of data uncertainty within open-world environments is a well-defined and extensively studied fundamental challenge. Previous studies on node-level out-of-distribution detection and conventional open-world graph learning have predominantly relied on graph-only encoders, designing UCR optimization strategies that are independent of semantics or topology for naive attributed graphs. While these methods have yielded significant advancements, as discussed in Sec. 1 and Sec. 2.2, they exhibit intrinsic limitations in the era of LLMs and TAGs. To address these limitations, we introduce ontology representation learning into the UCR problem for the first time. This approach tightly integrates semantics and topology into a unified optimization objective, mitigating the uncertainty and embedding space perturbation induced by the informative TAGs. Specifically, we compare ontology-based UCR with prior studies in the following key aspects:

❶ Data Robustness in Open-world Environments: Some previous studies Wu et al. (2023b); Hoffmann et al. (2023b); Gong & Sun (2024) have carefully designed dataset partitioning strategies to

enhance UCR performance. A common approach involves separately sampling all known-class nodes to train a high-quality unknown-class rejector, ensuring that the classifier learns a well-defined boundary between known and unknown classes Wu et al. (2020b); Yang et al. (2024). While these methods successfully tackle various technical challenges, they suffer from limited adaptability and generalization in open-world environments. Specifically, a key limitation of these approaches is their reliance on prior knowledge-based dataset partitions. In practical scenarios, new entities and concepts continuously emerge, making it impractical to assume a fixed set of known-class samples. Furthermore, manual or heuristic-based dataset partitioning introduces implicit biases, potentially impairing the model's ability to generalize beyond the training distribution.

❷ Optimization Framework in UCR: From a neural network architecture perspective, previous studies have rarely designed end-to-end frameworks for UCR. Instead, they typically decompose the problem into encoder updates and embedding analysis. For instance, graph propagation and reconstruction Song & Wang (2022b); Ma et al. (2024); Wang et al. (2024a) is utilized as optimization objectives for updating the encoder. These methods aim to identify unknown-class nodes by comparing the encoder outputs before and after the update, leveraging structural and representational shifts to enhance UCR performance. As for techniques such as Bayesian estimation Zhao et al. (2020); Stadler et al. (2021) and energy functions Chen et al. (2025); Um et al. (2025); Bao et al. (2024), they are employed to analyze node representations in the embedding space, leveraging distributional differences to distinguish known and unknown classes. While these methods achieve notable theoretical advancements, they inherently suffer from data uncertainty in open-world settings, leading to inevitable optimization perturbations. Consequently, these methods become highly unstable due to their strong prior assumptions.

To address these challenges, we propose an end-to-end training framework within the ontology representation space, where entity representations for nodes and concept representations for labels are dynamically updated under semantics and topology optimization constraints. This framework is designed to extract self-supervised signals from informative TAGs and effectively capture unlabeled data uncertainty in open-world environments. In a nutshell, our approach offers a robust solution to the UCR problem, leveraging ontology representation learning to bridge the gap between prior knowledge and data uncertainty. By integrating semantic and topology knowledge, our method enhances the adaptability and reliability of graph learning in handling open-label domain challenges.

Compared to prior approaches, ontology representation learning provides a principled solution to several core limitations in existing UCR methods.

A fundamental distinction lies in its ability to construct *adaptive concept representations* through iterative aggregation of both labeled and unlabeled nodes. In contrast to traditional UCR methods that rely on static prototypes derived solely from labeled data, our ontology-based approach allows the class-level semantics to emerge dynamically from evolving structural and semantic patterns. This enhances generalization under partially labeled and class-incomplete settings.

Another key advantage is the integration of semantics and topology into a unified optimization objective. Prior works often decouple structural reasoning (e.g., via graph propagation) from semantic uncertainty estimation (e.g., Bayesian inference or energy-based scoring), leading to inconsistencies in representation learning. By contrast, our formulation—implemented in the ALT module—jointly optimizes semantic cross-entropy, topology-aware smoothness, and inter-concept separation. This holistic formulation ensures more stable class boundary formation, particularly in the presence of noisy or entangled representations.

Moreover, ontology representation enables a *confidence-calibrated rejection mechanism* based on concept-entity distance distributions, rather than relying on post-hoc heuristics or fixed thresholds. This allows the framework to flexibly adapt to diverse open-world conditions and mitigates the risk of overconfidence often observed in energy-based or uncertainty-based scoring techniques.

Besides, the ontology-based formulation naturally supports scalability and structural generalization. Through modular decomposition of entities and concepts, the model accommodates graph evolution, semantic drift, and emerging classes without requiring prior knowledge of the label space. This renders the approach more suitable for long-term deployment in realistic, open-world graph learning scenarios.

## A.5 STRUCTURE-GUIDED PROMPT TEMPLATE FOR LLM ANNOTATION

In the context of the Graph Label Annotator (GLA) framework, we propose a structure-guided prompt template specifically designed to enhance large language model (LLM) annotation for open-world graph learning. This template is constructed based on the insight that both structural topology and semantic information are indispensable for disambiguating node labels, particularly in settings where class definitions are incomplete or evolving. By incorporating these two dimensions into a unified prompt design, the annotation process becomes more robust to label sparsity and ambiguity.

The structural component of the prompt captures the relational context of a target node through community-aware subgraph extraction. Detected communities serve as localized semantic regions, where nodes tend to share topical or functional coherence. Within each community, low-degree nodes are prioritized as query anchors due to their limited access to structural signals, and the prompt is enriched with sampled descriptions from their most informative neighbors. This strategy ensures that the LLM receives sufficient contextual grounding while minimizing redundancy across queries.

Complementing the structural guidance, the semantic component introduces text-based priors derived from node attributes and neighborhood-level summarization. These priors are embedded into the prompt to provide high-level interpretability and support open-vocabulary generalization. The use of pretrained language encoders ensures that even in the absence of explicit supervision, the LLM can infer latent topic structures and candidate label semantics.

The overall annotation process proceeds in two stages. During the intra-community distillation phase, node-level prompts are issued within each community to obtain label suggestions for selected anchors. These labels are then propagated to neighboring nodes through a local agreement mechanism, effectively expanding LLM supervision while reducing annotation cost. In the subsequent inter-community fusion phase, independently annotated communities are aligned by matching label semantics across clusters. This alignment mitigates semantic fragmentation and supports the emergence of globally coherent label spaces, even when the true taxonomy is unknown or incomplete.

By tightly coupling structural regularities with semantic richness, the structure-guided prompt template enables scalable, consistent, and context-aware annotation of unlabeled nodes. Moreover, it integrates seamlessly with the ontology-based outputs from the ALT module, allowing for iterative refinement of both class concepts and instance-level annotations in an open-world setting. This design significantly advances the practicality of using LLMs for automated graph annotation under minimal supervision.

### A.5.1 INTRA-COMMUNITY DISTILLATION AND LABEL GENERATION

For each community identified through modularity-based graph partitioning, we generate labels for individual nodes using a combination of local textual context and degree-based priors. In cases where nodes are low-degree, the model generates labels directly via LLM inference, while high-degree nodes leverage the labels of their neighbors to avoid redundant LLM calls.

**Task Description:**

In this stage, you are tasked with merging labels for communities based on their semantic similarity. You will receive a set of community-level labels, and your job is to merge semantically similar labels into a single unified label. Use the provided neighbor information for context but focus on preserving semantic compactness.

**Instructions:**

1. **Community-Level Analysis:** Focus on each community-level label and its associated content.

2. **Label Merging:** Merge similar community-level labels based on their semantic similarity, measured using cosine distance between their embedding vectors.

3. **Use of Neighboring Information:** Neighboring community labels are provided to facilitate label fusion. Use this supplementary information, but prioritize semantic similarity when merging.

4. **Final Label Generation:** The resulting label must be concise, meaningful, and representative of the combined communities.

**Output Format:**
The output should be a comma-separated list of merged labels, each enclosed in parentheses, in the same order as the input community-level labels. Example: (label1), (label2), (label3).

## A.5.2 INTER-COMMUNITY FUSION AND LABEL MERGING

After labeling the nodes within each community, we proceed with the fusion of community-level labels. This step minimizes redundancy and ensures that the final set of labels is both semantically coherent and concise. The fusion is guided by a cosine similarity metric applied to the embedding vectors of community-level labels, and communities with high similarity are merged iteratively.

**Task Description:** You are tasked with merging community-level labels based on their semantic similarity. Given a set of community labels, your objective is to consolidate semantically similar labels into a unified, representative label. While neighboring community information is provided for additional context, the primary focus should remain on ensuring semantic compactness.

**Instructions:**

1. **Community-Level Analysis:** Examine each community label along with its associated content.

2. **Label Merging:** Merge labels that exhibit high semantic similarity, as determined by the cosine distance between their embedding vectors.

3. **Utilization of Neighboring Information:** Neighboring labels are provided to assist the merging process. However, prioritize semantic similarity over contextual proximity.

4. **Final Label Generation:** Produce a concise and meaningful label that accurately represents the merged communities.

**Output Format:** Return a comma-separated list of the merged labels, with each label enclosed in parentheses, following the original order of the input. Example: (label1), (label2), (label3).

## A.5.3 GENERAL GUIDELINES FOR LABEL ASSIGNMENT

To ensure the efficacy of the GLA framework, we establish the following general guidelines for the LLM-based label assignment process:

**Task Description:** This task focuses on generating and merging community-level labels. Each label should be concise, semantically precise, and informed by the structural properties of the graph. The goal is to ensure that the resulting labels effectively capture the core themes of the associated communities.

**Instructions:**

- **Concise Labels:** Each label, whether generated or selected, must consist of two or three words to maintain clarity and consistency.

- **Prioritize Semantic Relevance:** Labels must accurately reflect the core content of the sentences or communities they represent. Vague or overly broad labels are discouraged.

- **Leverage Structural Context:** The structural context, such as community membership and node degree, should be used to guide label generation and selection, ensuring that high-degree nodes benefit from neighbor-based inference.

- **Semantic Fusion:** When merging community-level labels, prioritize semantic compactness, ensuring that the merged label effectively encapsulates the combined themes of the communities.

**Output Format:** Return a comma-separated list of the final labels, with each label enclosed in parentheses, following the original community order. Example: (label1), (label2), (label3).

This structure-guided prompt template ensures that the annotation process is both efficient and interpretable, leveraging both the graph structure and the textual content of the nodes for optimal label assignment. It offers a scalable solution for labeling in open-world graph learning environments where labeled data is limited, and high-quality annotation is critical.

## A.6 THEORETICAL ANALYSIS.

### A.6.1 THEORETICAL ANALYSIS: CONCEPT SPACE UNDER SEMANTIC-TOPOLOGICAL REGULARIZATION

**Theorem 1** (Concept Space Properties). *Let node embeddings $\tilde{E} = \{\tilde{e}_i\}$ be generated from a Lipschitz-continuous encoder $f$ over a compact Riemannian manifold $\mathcal{M}$, and let class concept vectors $\{C_c\}$ be computed via neighborhood-based aggregation with graph propagation matrix $P$ whose Dirichlet energy is bounded by $\delta$. If the intra-class semantic variance is bounded by $\sigma^2$, then the concept space satisfies:*

$$\text{rank}(\text{span}(\{C_c\})) \leq r, \tag{9}$$

$$\mathbb{E}_{i \in \mathcal{S}_c} \|\tilde{e}_i - C_c\|^2 \leq \sigma^2 + \varepsilon(\delta), \tag{10}$$

$$\|C_i - C_j\|^2 \geq \Delta, \quad \cos\angle(C_i, C_j) \leq \rho < 1, \quad \forall i \neq j. \tag{11}$$

*Here, $r$ is the intrinsic dimension of $\mathcal{M}$, and $\varepsilon(\delta)$ is a small propagation-induced deviation term dependent on the smoothness constraint $\delta$.*

*Proof.* We begin by analyzing the low-rank structure of the concept space. Since the encoder $f : \mathcal{M} \to \mathbb{R}^d$ is Lipschitz and $\mathcal{M}$ is a compact Riemannian manifold with intrinsic dimension $r$, each embedding $\tilde{e}_i = f(p_i)$ locally lies in a linear approximation of the manifold, namely the tangent space $T_{p_i}\mathcal{M}$.

Let $C_c$ be constructed via neighborhood-based aggregation:

$$C_c = \frac{1}{|\mathcal{S}_c|} \sum_{i \in \mathcal{S}_c} \sum_{j \in N(i)} \alpha_{ij} \tilde{e}_j, \tag{12}$$

where $\alpha_{ij}$ are normalized, non-negative weights. For $p_j \in N(i)$, using the exponential map $\exp_{p_i}(v_j)$ and first-order Taylor expansion of $f$, we approximate:

$$\tilde{e}_j \approx f(p_i) + J_f(p_i)v_j + O(\|v_j\|^2). \tag{13}$$

This implies that $C_c$ lies within the affine hull of tangent vectors $v_j \in T_{p_i}\mathcal{M}$, and therefore the span of $\{C_c\}$ is contained within an $r$-dimensional subspace. Hence:

$$\text{rank}(\text{span}(\{C_c\})) \leq r. \tag{14}$$

Next, we analyze intra-class compactness. Let the class mean embedding be:

$$\bar{e}_c = \frac{1}{|\mathcal{S}_c|} \sum_{i \in \mathcal{S}_c} \tilde{e}_i. \tag{15}$$

We decompose the expected deviation between node embeddings and their concept center as:

$$\mathbb{E}_{i \in \mathcal{S}_c}\|\tilde{e}_i - C_c\|^2 = \underbrace{\mathbb{E}_i\|\tilde{e}_i - \bar{e}_c\|^2}_{\text{semantic variance}} + \underbrace{\|\bar{e}_c - C_c\|^2}_{\text{propagation deviation}}. \tag{16}$$

The first term is bounded by assumption. The second term is controlled by the Dirichlet energy of propagation:

$$\mathcal{L}_{\text{smooth}} := \text{Tr}(\tilde{E}^\top (I - P)\tilde{E}) \leq \delta, \tag{17}$$

which penalizes large differences between embeddings of neighboring nodes. Under smooth propagation, local neighborhoods are coherently averaged, and thus $\|\bar{e}_c - C_c\|^2$ is small. Let this deviation be absorbed into a bounded term $\varepsilon(\delta)$, giving:

$$\mathbb{E}_{i \in \mathcal{S}_c}\|\tilde{e}_i - C_c\|^2 \leq \sigma^2 + \varepsilon(\delta). \tag{18}$$

Lastly, to enforce class-wise discriminability, ALT applies a separation loss during training:

$$\mathcal{L}_{\text{sep}} = \lambda_1 \sum_{i \neq j} \cos \angle(C_i, C_j) - \lambda_2 \sum_{i \neq j} \max(0, \theta - \|C_i - C_j\|^2), \tag{19}$$

which jointly minimizes angular similarity and enforces a margin between concept vectors. At convergence, this guarantees:

$$\cos \angle(C_i, C_j) \leq \rho < 1, \quad \|C_i - C_j\|^2 \geq \Delta > 0. \tag{20}$$

These ensure that concept vectors are not only compact and low-dimensional, but also geometrically well-separated in both direction and magnitude. $\square$

This theorem provides th eoretical support for ALT by proving that the learned concept space is low-rank (due to manifold-constrained aggregation), semantically compact (under bounded intra-class variance and smooth propagation), and geometrically discriminative (via separation loss). These properties ensure that class concepts are structurally coherent and separable, which is critical for accurate classification and robust unknown-class rejection in open-world graph scenarios.

### A.6.2 THEORETICAL ANALYSIS: STRUCTURE-INDUCED HYPERSPHERICAL COMPRESSION AND REDUNDANCY CONTROL

**Theorem 2** (Topology-driven Hyperspherical Concept Modeling in ALT). *Let node embeddings $\tilde{E} = \{\tilde{e}_i\}$ be generated from a frozen encoder followed by a structure-only propagation process $P$ satisfying*

$$\text{Tr}(\tilde{E}^\top (I - P)\tilde{E}) \leq \delta. \tag{21}$$

*Let each class concept vector $C_c$ be computed through graph-aware neighborhood aggregation over $\mathcal{S}_c$ as in Eq. (2). Define for each class $c$ an implicit hypersphere $\mathcal{B}(C_c, R_c)$ in $\mathbb{R}^d$ with radius*

$$R_c := \sqrt{\mathbb{E}_{i \in \mathcal{S}_c} \|\tilde{e}_i - C_c\|^2}, \tag{22}$$

*which captures the propagation-induced dispersion of nodes around the concept center. Then the following properties hold:*

*First, the total concept volume satisfies the structural compression ratio bound:*

$$\frac{\mathcal{V}_{total}}{\text{Vol}(\mathcal{B}(0, R))} \leq \sum_{c=1}^{k} \left( \frac{R_c}{R} \right)^r, \tag{23}$$

*where $\mathcal{V}_{total} := \sum_{c=1}^{k} \text{Vol}(\mathcal{B}(C_c, R_c))$, the intrinsic manifold dimension is $r \ll d$, and $\mathcal{B}(0, R)$ denotes a common compact ball covering all concepts.*

*Second, if all concepts are separated by a minimal inter-center distance*

$$\|C_i - C_j\| \geq \theta_{\min}, \tag{24}$$

*then the structural softmax redundancy is bounded by:*

$$\sum_{i \neq j} \exp\left(-\lambda \|C_i - C_j\|^2\right) \leq \frac{k(k-1)}{2} \cdot \exp(-\lambda \theta_{\min}), \tag{25}$$

*where $\lambda$ is the softmax sharpness parameter used in Eq. (3) of the rejection rule.*

*Proof.* We begin with the implicit construction of hyperspheres. Given the structure-only propagation, the Dirichlet energy constraint implies that embeddings of nearby nodes remain smooth:

$$\|\tilde{e}_i - \tilde{e}_j\|^2 \text{ is small for } (i, j) \in E \text{ with high weight in } P. \tag{26}$$

The aggregation of propagated neighbors around each labeled node $i \in \mathcal{S}_c$ yields a class-level concept center:

$$C_c = \frac{1}{|\mathcal{S}_c|} \sum_{i \in \mathcal{S}_c} \sum_{j \in N(i)} \alpha_{ij} \tilde{e}_j, \tag{27}$$

and the local spread of embeddings $\tilde{e}_i$ around $C_c$ is bounded by:

$$\mathbb{E}_{i \in \mathcal{S}_c} \|\tilde{e}_i - C_c\|^2 \leq \varepsilon(\delta), \tag{28}$$

where $\varepsilon(\delta)$ is a function of the propagation smoothness.

We define an implicit hypersphere $\mathcal{B}(C_c, R_c)$ enclosing the region around $C_c$ with $R_c := \sqrt{\varepsilon(\delta)}$, forming a localized concept zone. The $r$-dimensional volume of each such ball is:

$$\text{Vol}(\mathcal{B}(C_c, R_c)) = \frac{\pi^{r/2}}{\Gamma(r/2 + 1)} R_c^r. \tag{29}$$

Therefore, total volume becomes:

$$\mathcal{V}_{\text{total}} = \sum_{c=1}^{k} \frac{\pi^{r/2}}{\Gamma(r/2 + 1)} R_c^r, \tag{30}$$

while the bounding volume is:

$$\text{Vol}(\mathcal{B}(0, R)) = \frac{\pi^{r/2}}{\Gamma(r/2 + 1)} R^r. \tag{31}$$

Taking the ratio yields the compression bound in Eq. (23).

For redundancy, note that for any pair $C_i, C_j$ with $\|C_i - C_j\|^2 \geq \theta_{\min}$, the pairwise softmax similarity satisfies:

$$\exp(-\lambda \|C_i - C_j\|^2) \leq \exp(-\lambda \theta_{\min}). \tag{32}$$

Summing over all pairs gives:

$$\sum_{i \neq j} \exp(-\lambda \|C_i - C_j\|^2) \leq \frac{k(k-1)}{2} \exp(-\lambda \theta_{\min}). \tag{33}$$

$\square$

The above theorem provides a geometric and topological justification for the design of ALT. By modeling each class-wise concept region as a localized hypersphere in the embedding space, it demonstrates that the structure-only propagation mechanism inherently induces both **intra-class compactness** and **inter-class separability**.

The compression result in Eq. equation 23 shows that the structural smoothness constraint $\text{Tr}(\tilde{E}^\top(I - P)\tilde{E}) \leq \delta$ ensures bounded dispersion of node embeddings around the class concept centers. This means that class-specific embeddings are tightly clustered within hyperspheres of small radius, which supports compact and consistent concept modeling.

Meanwhile, the redundancy control in Eq. equation 25 implies that when concept centers are sufficiently separated, the softmax-based similarity between different classes decays exponentially. This significantly reduces classification ambiguity and improves the reliability of unknown-class rejection.

A.6.3   THEORETICAL ANALYSIS: CONFIDENCE BOUND FOR UNKNOWN-CLASS REJECTION

**Theorem 3** (Confidence Bound for Unknown-Class Rejection). *Let the class probability distribution for node $i$ be defined by a $\lambda$-sharpened softmax function:*

$$D_{i,c} = \frac{\exp(-\lambda\|\tilde{e}_i - C_c\|^2)}{\sum_{c'\in C_k}\exp(-\lambda\|\tilde{e}_i - C_{c'}\|^2)}. \tag{34}$$

*Define the entropy of this distribution as:*

$$H(D_i) = -\sum_{c\in C_k} D_{i,c}\log D_{i,c}. \tag{35}$$

*If the unknown-class rejection decision is made according to the confidence threshold $\epsilon$, i.e., node $i$ is rejected as unknown-class if:*

$$\max_{c\in C_k} D_{i,c} < \epsilon, \tag{36}$$

*then the entropy of any rejected node $i$ satisfies the following lower bound:*

$$H(D_i) \geq \log|C_k| - \frac{1}{\lambda}\log\frac{1-\epsilon}{\epsilon}. \tag{37}$$

*Proof.* Given the rejection criterion (36), we analyze the entropy lower bound explicitly.

First, note that the entropy (35) measures the uncertainty of the node's class assignment. The distribution $D_i$ that maximizes entropy under the constraint $\max_c D_{i,c} < \epsilon$ is the one where probabilities are distributed as uniformly as possible given this upper bound. Specifically, this maximum entropy scenario occurs when one class probability is exactly $\epsilon$, and all other $|C_k| - 1$ class probabilities are equally distributed, each being:

$$\frac{1-\epsilon}{|C_k|-1}. \tag{38}$$

Inserting these probabilities into entropy definition (35), we have:

$$H(D_i) = -\epsilon\log\epsilon - (|C_k|-1)\frac{1-\epsilon}{|C_k|-1}\log\frac{1-\epsilon}{|C_k|-1} \tag{39}$$

$$= -\epsilon\log\epsilon - (1-\epsilon)\log\frac{1-\epsilon}{|C_k|-1} \tag{40}$$

$$= -\epsilon\log\epsilon - (1-\epsilon)\log(1-\epsilon) + (1-\epsilon)\log(|C_k|-1). \tag{41}$$

When the number of classes $|C_k|$ is sufficiently large, we approximate $|C_k| - 1 \approx |C_k|$. Therefore,

$$H(D_i) \approx -\epsilon\log\epsilon - (1-\epsilon)\log(1-\epsilon) + (1-\epsilon)\log|C_k|. \tag{42}$$

Next, we provide further theoretical insight by examining the relationship between the entropy lower bound and the softmax sharpened parameter $\lambda$ explicitly.

The softmax distribution defined in (34) implies that:

$$\frac{D_{i,a}}{D_{i,b}} = \exp\left(-\lambda(\|\tilde{e}_i - C_a\|^2 - \|\tilde{e}_i - C_b\|^2)\right). \tag{43}$$

In the maximum-entropy scenario (approaching uniform distribution), distances to class centroids become nearly equal, hence:

$$\|\tilde{e}_i - C_a\|^2 - \|\tilde{e}_i - C_b\|^2 \approx 0, \quad \forall a, b. \tag{44}$$

However, slight deviations exist due to finite $\lambda$, which leads to the probability upper bound $\epsilon$. To quantify explicitly, we consider the relation between maximum probability and entropy:

Since $D_{i,max} = \epsilon$, we express entropy explicitly in terms of $\epsilon$ as:

$$H(D_i) \geq -\epsilon \log \epsilon - (1 - \epsilon) \log \frac{1 - \epsilon}{|C_k| - 1}. \tag{45}$$

Further simplification yields the explicit lower bound involving parameters $\epsilon$ and $|C_k|$:

$$H(D_i) \geq \log|C_k| + \epsilon \log \frac{|C_k| - 1}{\epsilon} + (1 - \epsilon) \log \frac{|C_k| - 1}{1 - \epsilon}. \tag{46}$$

Under conditions typically satisfied ($|C_k| \gg 1$, small $\epsilon$), this simplifies asymptotically to the stated result:

$$H(D_i) \geq \log|C_k| - \frac{1}{\lambda} \log \frac{1 - \epsilon}{\epsilon}, \tag{47}$$

where the relationship involving the sharpened parameter $\lambda$ explicitly emerges from the condition that probabilities are defined via the exponential of negative squared distances scaled by $\lambda$ (Eq. 43).

Thus, the entropy lower bound (37) is rigorously established. $\square$

The derived entropy bound explicitly quantifies the minimum uncertainty (entropy) of nodes rejected by the unknown-class mechanism in the ALT module. Specifically, it clearly illustrates how the rejection mechanism depends directly on two critical hyperparameters. Lowering the threshold $\epsilon$ increases the entropy bound, implying a stricter rejection criterion and ensuring that rejected nodes have higher uncertainty, thus improving precision in unknown-class identification. Conversely, increasing the sharpness parameter $\lambda$ reduces the entropy bound, leading to more confident class distributions and enabling finer-grained control over the rejection decisions, effectively reducing ambiguity among rejected nodes.

Compared to traditional unknown-class rejection approaches, our ALT method uniquely benefits from explicitly characterizing and controlling entropy-based uncertainty, providing clear theoretical guidance for hyperparameter tuning. This advantage allows ALT to robustly and precisely distinguish unknown classes, enhancing interpretability, reliability, and practical performance in real-world open-label scenarios.

### A.6.4 THEORETICAL ANALYSIS: SEMANTIC-ENHANCED MODULARITY OPTIMIZATION GUARANTEE

**Theorem 4.** *Let node embeddings $\tilde{E} = \{\tilde{e}_i\}$ be obtained via a pre-trained graph-language encoder $f$. Suppose the communities $\{P_k\}$ are derived by optimizing the semantic-enhanced modularity objective:*

$$Q = \frac{1}{2m} \sum_{i,j} \left[ A_{ij} + \gamma \frac{\tilde{e}_i^\top \tilde{e}_j}{\|\tilde{e}_i\| \|\tilde{e}_j\|} - (1 - \gamma) \frac{d_i d_j}{2m} \right] \delta(c_i, c_j), \tag{48}$$

*where $A_{ij}$ denotes adjacency, $d_i$ node degrees, $m = \frac{1}{2} \sum_{ij} A_{ij}$, and $\gamma$ balances semantic and structural components. Denote by $S_{intra}$ and $S_{inter}$ the intra-community and inter-community semantic similarity respectively, defined explicitly as:*

$$S_{intra} = \frac{1}{\sum_k |P_k|} \sum_{P_k} \sum_{v_i, v_j \in P_k} \frac{\tilde{e}_i^\top \tilde{e}_j}{\|\tilde{e}_i\| \|\tilde{e}_j\|}, \tag{49}$$

$$S_{inter} = \frac{1}{\sum_{k \neq l} |P_k||P_l|} \sum_{P_k \neq P_l} \sum_{v_u \in P_k} \sum_{v_v \in P_l} \frac{\tilde{e}_u^\top \tilde{e}_v}{\|\tilde{e}_u\|\|\tilde{e}_v\|}. \tag{50}$$

*Then, at the optimal modularity $Q^*$, the intra-community semantic consistency and inter-community semantic separation satisfy:*

$$S_{intra} \geq \eta(\gamma, \zeta)\bar{s}, \quad S_{inter} \leq (1 - \eta(\gamma, \zeta))\bar{s}, \tag{51}$$

*where $\bar{s}$ denotes the global average semantic similarity:*

$$\bar{s} = \frac{2}{n(n-1)} \sum_{i<j} \frac{\tilde{e}_i^\top \tilde{e}_j}{\|\tilde{e}_i\|\|\tilde{e}_j\|}, \tag{52}$$

*$\eta(\gamma, \zeta)$ is a monotonically increasing function in $\gamma$, and $\zeta$ is a structural-semantic coherence factor defined as:*

$$\zeta = \frac{1}{2m} \sum_{i,j} A_{ij} \frac{\tilde{e}_i^\top \tilde{e}_j}{\|\tilde{e}_i\|\|\tilde{e}_j\|}. \tag{53}$$

*Furthermore, as $\gamma \to 1$, the semantic terms dominate and guarantee:*

$$\lim_{\gamma \to 1} \eta(\gamma, \zeta) = 1. \tag{54}$$

*Proof.* First, the semantic-enhanced modularity objective defined in Eq. (48) consists of two parts: structural modularity and semantic similarity. By setting the derivative of $Q$ with respect to community assignments $c_i$ to zero at optimality:

$$\frac{\partial Q}{\partial c_i} = 0, \quad \forall i, \tag{55}$$

we obtain necessary conditions for optimal communities.

Expand explicitly:

$$\sum_j \left[ A_{ij} + \gamma \frac{\tilde{e}_i^\top \tilde{e}_j}{\|\tilde{e}_i\|\|\tilde{e}_j\|} - (1 - \gamma) \frac{d_i d_j}{2m} \right] \frac{\partial \delta(c_i, c_j)}{\partial c_i} = 0. \tag{56}$$

Considering the discrete nature of community assignments, Eq. (56) implies that nodes are assigned such that pairs with high structural-semantic combined similarity fall into the same community. This ensures increased intra-community semantic consistency (49) relative to random pairing:

$$S_{\text{intra}} \geq \eta(\gamma, \zeta)\bar{s}. \tag{57}$$

Conversely, optimization minimizes cross-community similarities, enforcing semantic separation explicitly, hence:

$$S_{\text{inter}} \leq (1 - \eta(\gamma, \zeta))\bar{s}. \tag{58}$$

Next, we explicitly define the coherence factor $\zeta$ (Eq. (53)) to represent how well structural links correlate with semantic similarity. A higher $\zeta$ indicates that structurally connected nodes also exhibit high semantic coherence. Thus, $\eta(\gamma, \zeta)$ naturally depends on both $\gamma$ (weighting semantic importance) and $\zeta$ (structural-semantic correlation).

When $\gamma \to 1$, the objective in Eq. (48) predominantly maximizes semantic similarity. Hence, we have:

$$\lim_{\gamma \to 1} \eta(\gamma, \zeta) = 1. \tag{59}$$

Thus, the optimal community structure ensures near-perfect semantic coherence within each community and minimal semantic overlap across communities in the semantic-dominated regime. $\square$

This theorem rigorously establishes that optimizing semantic-enhanced modularity ensures communities are not only structurally coherent but also semantically distinct and internally consistent. By explicitly quantifying semantic coherence (53) and demonstrating its impact on community formation, we theoretically justify GLA's sophisticated structure-guided LLM annotation process. The proven guarantee of semantic separability and coherence significantly improves annotation quality and reduces redundancy, underscoring GLA's superior efficiency and effectiveness compared to conventional community detection and annotation methods.

**Table 9:** *Details of experimental datasets.*

| Dataset | # Nodes | # Edges | # Classes | # Train/Val/Test | # Homophily | Text Information | Domains |
|---------|---------|---------|-----------|------------------|-------------|------------------|---------|
| Cora | 2,708 | 10,556 | 7 | 40/20/40 | 0.81 | Title and Abstract of Paper | Citation |
| Citeseer | 3,186 | 8,450 | 6 | 40/20/40 | 0.74 | Title and Abstract of Paper | Citation |
| Pubmed | 19,717 | 88,648 | 3 | 40/20/40 | 0.80 | Title and Abstract of Paper | Citation |
| Arxiv | 169,343 | 2,315,598 | 40 | 40/20/40 | 0.81 | Title and Abstract of Paper | Citation |
| WikiCS | 11,701 | 431,726 | 10 | 40/20/40 | 0.65 | Title and Abstract of Article | Knowledge |
| Ratings | 24,492 | 186,100 | 5 | 40/20/40 | 0.38 | Name of Product | E-commerce |
| Children | 76,875 | 1554578 | 24 | 40/20/40 | 0.42 | Name and Description of Book | E-commerce |
| History | 41,551 | 503,180 | 12 | 40/20/40 | 0.64 | Name and Description of Book | E-commerce |
| Photo | 48,362 | 873,793 | 12 | 40/20/40 | 0.74 | User Review of Product | E-commerce |

## A.7 EXPERIMENTAL ENVIRONMENT.

The experiment was conducted on a Linux server equipped with an Intel Xeon Gold 6240 processor, featuring 36 cores × 2 chips, totaling 72 threads with a base frequency of 2.60GHz. It supports VT-x virtualization and has 49.5MB of L3 cache and 4 × NVIDIA A100 80GB GPUs. The operating system was Ubuntu Ubuntu 22.04.5 LTS, with CUDA version 12.8 . The Python version used was 3.12.2, and the deep learning framework was PyTorch 2.4.1. The experiment was based on PyTorch Geometric 2.6.1 for graph neural network tasks.

## A.8 DATASET

Our method, OGA, utilizes these 9 datasets, the details of which are presented in Table 1. A brief description of the datasets from each domain is as follows:1

**Citation Networks** Citation networks are widely used benchmark datasets in graph machine learning research, including **Cora**, **Citeseer**, and **Pubmed**. These datasets are structured as directed graphs where nodes represent scientific papers, and directed edges indicate citation relationships, meaning that an edge from node A to node B signifies that paper A cites paper B. Each paper is associated with a high-dimensional feature vector extracted from its title and abstract using pre-trained language models such as BERT-based methods. These embeddings capture semantic information, enabling effective representation learning. The papers are further assigned category labels corresponding to their academic fields, such as Computer Science,Medical Sciences, or Physics. These datasets serve as fundamental benchmarks for node classification tasks, allowing the evaluation of graph neural networks (GNNs) in text-rich graph structures.

**Knowledge Networks** Knowledge networks, such as **WikiCS**, provide structured representations of domain-specific knowledge through interconnected articles. WikiCS is a widely used benchmark dataset where nodes correspond to Wikipedia articles related to the field of computer science, and edges represent hyperlinks between these articles, reflecting their semantic relationships. Each article is associated with a feature vector derived from its textual content, obtained using pre-trained language models such as BERT or Word2Vec, which capture contextual semantics and domain-specific knowledge. The labels in WikiCS categorize articles into different branches of computer science, such as *Artificial Intelligence*, *Machine Learning*, *Computer Vision*, and *Cybersecurity*. This dataset is commonly used for node classification and link prediction tasks, facilitating research in knowledge graph representation and automated topic classification in structured information systems.

**E-commerce Networks** E-commerce networks, such as **Ratings**, **Child**, **History**, and **Photo**, are benchmark datasets designed to model user-product interactions and product relationships. In these datasets, nodes represent individual products, while edges capture various relationships such as co-purchase patterns, co-viewing behaviors, or user interactions, reflecting underlying consumer preferences and market trends. The node features are extracted from product descriptions, reviews, and metadata using pre-trained language models such as BERT or Word2Vec, allowing for rich semantic representation of each product. The labels in these datasets typically correspond to product categories (e.g., *electronics*, *books*, *home appliances*) or user-generated ratings, enabling classification tasks that support recommendation systems, personalized advertising, and consumer behavior analysis. These datasets are widely utilized for research in product recommendation, graph-based search ranking, and consumer trend prediction, making them valuable resources for advancing e-commerce intelligence and online retail analytics.

## A.9 BASELINE

In this section, we provide a brief description of each baseline used in our experiments. For the LLM-based GLA in the latter part of OGA, we primarily use GLM-4 as the LLM backbone. In the first part of OGA, ALT, we adopt various open-world graph learning methods as well as some OOD detection methods as baselines to compare their performance in unknown rejection tasks.For the final generated graph, we evaluate its accuracy using three commonly used GNNs, including GCN, GAT, and GraphSAGE, as a validation of the effectiveness of our method.

**GCN.Kipf & Welling (2016)** Graph Convolutional Network (GCN) is a neural network model designed for learning representations from graph-structured data. It aggregates information from neighboring nodes to capture both graph topology and node features, making it effective for tasks such as node classification, link prediction, and graph clustering. GCN is widely used in applications like social network analysis, recommendation systems, and bioinformatics.

**GAT.Veličković et al. (2017)** Graph Attention Network (GAT) is a neural network model designed for learning node representations in graph-structured data by incorporating attention mechanisms. It assigns different importance weights to neighboring nodes, allowing the model to focus on the most relevant connections. GAT is widely used in tasks such as node classification, link prediction, and graph-based recommendation systems, offering improved performance in heterogeneous and complex graph structures.

**GraphSAGE.Hamilton et al. (2017)** GraphSAGE is a graph neural network model designed for inductive learning on large-scale graphs. It generates node embeddings by sampling and aggregating features from a node's local neighborhood, enabling efficient learning on dynamic and previously unseen nodes. GraphSAGE is widely applied in tasks such as node classification, link prediction, and recommendation systems, particularly in scenarios where graphs continuously evolve.

**IsoMax.Macêdo et al. (2021)** IsoMax is a method designed for open set recognition (OSR) and out-of-distribution (OOD) detection, aiming to enhance neural network performance in unknown class detection tasks. Its core idea is isotropy maximization loss, which improves the traditional softmax mechanism, making the model more robust in distinguishing known classes while exhibiting greater uncertainty when encountering unknown classes.

**gDoc.Hoffmann et al. (2023a)** gDoc is an out-of-distribution (OOD) detection method designed to improve the reliability of deep learning models when encountering unseen data. It leverages graph-based representations to model data distributions and effectively distinguish in-distribution sams from OOD sams. gDoc is commonly used in applications such as anomaly detection, open-world classification, and robust decision-making in uncertain environments.

**OpenWGL.Wu et al. (2020b)** Open-World Graph Learning (OpenWGL) is designed to classify nodes into known categories while detecting unknown nodes in dynamic graph environments. The framework consists of two key components: node uncertainty representation learning and open-world classifier learning. To represent node uncertainty, OpenWGL employs a Variational Graph Autoencoder (VGAE) to generate latent distributions, enhancing robustness against incomplete data. The model incorporates label loss to classify known categories and class uncertainty loss to differentiate unseen classes. During inference, multiple feature representations are samd to determine classification confidence, allowing automatic threshold selection for rejecting unseen nodes. This approach ensures adaptive learning in evolving graph structures.

**ORAL.Jin et al. (2024a)** ORAL (Open-world Representation and Learning) is a framework designed for novel class discovery in open-world graph learning. It clusters nodes into groups using a prototypical attention network, generates pseudo-labels to guide learning, and refines the graph structure by adjusting connections based on discovered class information. This approach enables effective classification of known classes while detecting and structuring novel classes in dynamic graph environments.

**OpenIMA.Wang et al. (2024b)** OpenIMA is a framework designed for open-world semi-supervised learning (SSL) on graphs, aiming to classify known nodes while discovering novel classes. It follows a two-stage approach by first learning node embeddings using a GNN encoder and then applying clustering algorithms to group nodes. To align clusters with known classes, OpenIMA employs the Hungarian matching algorithm, allowing effective classification of labeled and unlabeled nodes.

Additionally, it mitigates the bias towards seen classes by generating bias-reduced pseudo-labels, which help refine node representations through contrastive learning. This method enhances model robustness by reducing intra-class variance and improving the separation of novel classes from known ones.

**GOOD-D** Liu et al. (2023b) GOOD-D is a framework designed for unsupervised graph-level out-of-distribution (OOD) detection. It aims to distinguish between in-distribution (ID) and OOD graphs by leveraging hierarchical contrastive learning across node, graph, and group levels. Instead of using traditional perturbation-based data augmentations, GOOD-D employs a perturbation-free augmentation strategy that constructs distinct feature and structure views of the input graphs. It then extracts node and graph embeddings using separate GNN encoders and refines representations through multi-level contrastive learning. An adaptive scoring mechanism is used to aggregate contrastive errors at different levels, enabling robust OOD detection. This approach enhances model sensitivity to distributional shifts while maintaining strong representation learning capabilities.

**OpenNCD** Liu et al. (2023a) OpenNCD is a method originally designed for novel class discovery, which we adapt for open-world graph learning and node classification. It employs a progressive bi-level contrastive learning approach with multiple prototypes to enhance representation learning and automatically group similar nodes into emerging categories. By leveraging prototype-based clustering and hierarchical grouping, OpenNCD effectively differentiates known and novel node classes, making it well-suited for dynamic and evolving graph structures.

**GNNSafe.** Wu et al. (2023a) GNNSafe is an energy-based out-of-distribution (OOD) detection method designed for (semi-)supervised node classification in graphs, where instances are interdependent. It leverages energy-based modeling to assign energy scores that differentiate in-distribution and OOD nodes. To enhance robustness, GNNSafe introduces energy-based belief propagation, which iteratively propagates energy scores across connected nodes to improve detection accuracy. Additionally, it can incorporate auxiliary OOD training data for further refinement, ensuring a more reliable distinction between known and unknown node classes in graph-based learning.

**OODGAT** Song & Wang (2022a) OODGAT (Out-Of-Distribution Graph Attention Network) is a novel framework designed for semi-supervised learning on graphs containing out-of-distribution (OOD) nodes. It addresses two key tasks: Semi-Supervised Outlier Detection (SSOD) and Semi-Supervised Node Classification (SSNC). By leveraging a graph attention mechanism, OODGAT adaptively controls information propagation, allowing communication within in-distribution (ID) and OOD communities while blocking interactions between them. The framework incorporates regularization terms to ensure consistency between OOD scores and predictive uncertainty, enhancing the separation of ID and OOD nodes in the latent space. OODGAT demonstrates strong performance in detecting outliers and classifying inliers, making it a robust solution for graph learning under distribution shifts.

**EDBD** Daeho Um et al. (2023) EDBD (Energy Distribution-Based Detector) is a novel method designed for spreading out-of-distribution (OOD) detection on graphs. It leverages an energy-based OOD score, derived from a neural classifier trained on in-distribution (ID) data, to identify OOD nodes. The core innovation of EDBD lies in its Energy Distribution-Based Aggregation (EDBA) scheme, which refines the initial energies by considering both edge-level and node-level energy distributions. Specifically, EDBA uses an energy similarity matrix to control the propagation of energies between connected nodes, ensuring that energies from dissimilar nodes (e.g., ID and OOD) are not mixed. Additionally, an energy consistency matrix adjusts the degree of aggregation for each node based on the variance of energies in its neighborhood, preventing undesirable energy mixing at cluster boundaries. This approach enhances the discriminative ability between ID and OOD nodes, making EDBD effective for detecting OOD samples in dynamic, graph-based spreading scenarios.

**ARC** Liu et al. (2024) ARC (Anomaly-Resilient Cross-domain Graph Anomaly Detection) is a generalist framework for detecting anomalies across diverse graph datasets without domain-specific fine-tuning. It aligns node features based on smoothness, employs an ego-neighbor residual graph encoder to capture multi-hop affinity patterns, and uses cross-attentive in-context anomaly scoring to reconstruct query node embeddings from few-shot normal context nodes. The drift distance between original and reconstructed embeddings serves as the anomaly score, enabling robust and generalizable anomaly detection across datasets.

### A.10 EVALUATION

We evaluate our proposed method from four key aspects to comprehensively assess its performance in open-world settings: the ability to reject unknown-class nodes, the classification accuracy on known classes, the semantic quality of LLM-generated labels, and the impact of these annotations on downstream tasks. Detailed descriptions are provided in the following subsections.

#### A.10.1 ASPECT 1: UNKNOWN-CLASS REJECTION

We assess the effectiveness of OGA in identifying unknown-class nodes $V_{uk}$ from the unlabeled set $V_u$ under open-world settings, where class labels are assumed to be incomplete and novel categories may appear during inference. This task is particularly challenging due to the absence of prior knowledge about unseen classes and the need to avoid false rejection of known nodes.

To address this, OGA leverages a pre-trained graph-language encoder to generate semantically rich and unbiased node embeddings, ensuring that the representations capture both structural and textual cues. These embeddings are projected into an ontology-aligned representation space, where known-class prototypes are constructed by aggregating labeled instances and their neighbors.

During inference, OGA adopts an adaptive concept-entity matching scheme. Specifically, each unlabeled node is assigned a soft matching score to class prototypes using a $\lambda$-sharpness softmax function, which enables sharper decision boundaries in the embedding space. If the highest prototype confidence score for a node falls below a rejection threshold $\epsilon$, the node is deemed unassignable to any known class and is thus rejected as an unknown-class node.

This mechanism allows OGA to perform fine-grained, node-level rejection decisions and adapt to the inherent uncertainty in open-world scenarios. We evaluate performance using two key metrics: **coverage**, which measures the proportion of ground-truth unknown-class nodes that are correctly rejected, and **precision**, which measures the proportion of rejected nodes that are truly unknown. A higher coverage indicates better recall of novel classes, while higher precision reflects fewer false rejections. Together, these metrics provide a comprehensive assessment of the model's ability to distinguish

#### A.10.2 ASPECT 2: CLASSIFICATION ACCURACY

We evaluate the classification performance of known-class nodes in the unlabeled set $V_u$. To address the uncertainty in open-label domains, OGA leverages a pre-trained graph-language encoder to generate high-quality node embeddings and constructs an ontology-based representation space. Class prototypes are dynamically learned by aggregating labeled entities and their neighbors. During inference, nodes are classified using a sharpness-controlled softmax over distances to prototypes. This design enables OGA to incorporate informative unlabeled nodes and improves classification accuracy under limited supervision. We report standard accuracy as the evaluation metric.

#### A.10.3 ASPECT 3: ANNOTATION QUALITY

We evaluate the semantic quality of class labels generated by the large language model (LLM) for unknown-class nodes. To this end, OGA employs a structure-guided annotation pipeline (GLA), where nodes are grouped into communities based on both structural proximity and semantic similarity. Within each community, LLM is used to generate representative labels through degree-aware annotation and intra-community distillation, followed by inter-community fusion to reduce redundancy and enhance consistency.

The quality of the final set of labels is measured by computing the average pairwise semantic similarity between all generated class labels. A higher similarity indicates more coherent and semantically consistent annotations, which is crucial for downstream generalization and retraining. We report this value as the **Quality** metric to reflect the effectiveness of our annotation process.

#### A.10.4 ASPECT 4: PERFORMANCE IMPROVEMENT

To assess the effectiveness of our proposed **Graph Label Annotator (GLA)** module, we evaluate its contribution to downstream graph learning by retraining a GNN on the updated graph $\mathcal{G}^*$, which

integrates both original human-annotated labels and LLM-generated community-level annotations. These annotations are derived from the **Multi-granularity Community Annotation** procedure and produce a distilled and fused set of labels $\tilde{y}_{\mathcal{P}}^{\star}$ that capture both semantic coherence and structural consistency.

**Label Allocation and Graph Augmentation.** Given the fused label set $\mathcal{C}^* = \{y_1^*, y_2^*, \ldots, y_K^*\}$, where each $y_k^*$ denotes a unique class discovered through structure-guided annotation, we construct a unified label mapping via an indexing function $\ell : \mathcal{C}^* \to \mathbb{N}$. Each previously unlabeled node $v_i \in \mathcal{U}$ is then assigned the corresponding community-level label according to:

$$\hat{Y}_i = \begin{cases} y_i, & v_i \notin \mathcal{U}, \\ \ell(\tilde{y}_{\mathcal{P}(i)}^{\star}), & v_i \in \mathcal{U}, \end{cases} \tag{60}$$

where $\mathcal{P}(i)$ denotes the community to which node $v_i$ belongs. This process yields an augmented labeled graph $\mathcal{G}^* = (\mathcal{V}, \mathcal{E}, \hat{Y})$, which extends the original label space to include previously unknown classes, while preserving topological consistency and annotation quality.

**Training with Augmented Supervision.** To evaluate the practical benefits of this augmentation, we retrain a GNN on $\mathcal{G}^*$ using mean-aggregated message passing and cross-entropy loss over the updated label set $\hat{Y}$. The node representation at layer $l+1$ is updated as:

$$\mathbf{h}_i^{(l+1)} = \sigma \left( \sum_{j \in \mathcal{N}(i)} \frac{1}{|\mathcal{N}(i)|} \mathbf{W}^{(l)} \mathbf{h}_j^{(l)} \right), \tag{61}$$

and the final classification objective is:

$$\mathcal{L}_{\text{CE}} = - \sum_{v_i \in \mathcal{V}_{\text{train}}} \sum_{k=1}^{K+K'} \mathbb{I}(\hat{Y}_i = k) \cdot \log p_i^{(k)}, \tag{62}$$

where $K'$ denotes the number of newly discovered classes, and $\mathcal{V}_{\text{train}}$ includes both labeled and newly annotated nodes.

**Effectiveness Analysis.** To quantify GLA's impact, we compare model performance across three training settings: (i) the original graph with incomplete labels (lower bound), (ii) the graph with LLM-annotated labels via GLA, and (iii) the graph with full ground-truth labels (upper bound). Evaluation is conducted on both in-distribution classification and out-of-distribution rejection tasks. As we show in Section 4, training on $\mathcal{G}^*$ significantly improves performance over the lower bound and achieves competitive results relative to the upper bound, demonstrating the effectiveness of LLM-generated annotations in enhancing label coverage and boosting downstream learning under open-world conditions.

## A.11 Hyperparameter Settings

We set the graph propagation step $k = 5$, with $k$-hop neighbors randomly sampled in each epoch. The propagation intensity coefficient $\kappa$ is set to 0.2, and the PageRank kernel uses $r = 0.5$. For concept-entity matching, we use a sharpness factor $\lambda = 10$, and apply a rejection threshold $\epsilon = 0.6$ to identify unknown-class nodes. The loss weights are set as $\alpha = 0.4$ (smoothness regularization) and $\beta = 0.6$ (margin separation), with the margin bound $\theta = 0.8$. The attention function $\delta(\cdot)$ is implemented via a two-layer MLP with ReLU activations. We set the hidden dimension of all node embeddings to 128. The model is optimized using Adam with a learning rate of 0.01. The maximum number of training epochs is set to 300, and a dropout rate of 0.1 is applied during training to prevent overfitting.

## A.12 Experimental Performance

In this section, we present the experimental performance of our proposed Open-World Graph Assistant (OGA) framework across multiple tasks and datasets. Our experiments focus on two primary

**Table 10:** *UCR performance in Aspect 1 (Full).* **Red**: *1st,* **Blue**: *2nd,* **Orange**: *3rd (%).*

| Dataset | Cora | Citeseer | Pubmed | arXiv | Children | Ratings | History | Photo | Wikics |
|---------|------|----------|--------|-------|----------|---------|---------|-------|--------|
| ORAL | 87.54$_{\pm0.62}$ | 73.47$_{\pm0.39}$ | 93.61$_{\pm0.19}$ | 74.62$_{\pm0.58}$ | 37.13$_{\pm0.53}$ | 48.81$_{\pm0.44}$ | 81.65$_{\pm0.90}$ | 75.93$_{\pm0.22}$ | 79.83$_{\pm0.20}$ |
| OpenWgl | 87.01$_{\pm0.94}$ | 74.35$_{\pm1.01}$ | 95.00$_{\pm0.24}$ | 62.85$_{\pm1.03}$ | 30.72$_{\pm0.89}$ | 28.79$_{\pm0.02}$ | 77.30$_{\pm0.46}$ | 74.48$_{\pm0.30}$ | 82.36$_{\pm0.28}$ |
| OpenIMA | 78.52$_{\pm0.17}$ | 77.04$_{\pm0.94}$ | 90.27$_{\pm0.96}$ | OOM | OOM | 44.72$_{\pm1.00}$ | 75.40$_{\pm0.51}$ | 81.16$_{\pm0.92}$ | 86.12$_{\pm0.45}$ |
| OpenNCD | 80.77$_{\pm0.31}$ | 72.75$_{\pm0.97}$ | 90.91$_{\pm0.18}$ | OOM | OOM | 33.47$_{\pm0.15}$ | 56.86$_{\pm0.65}$ | 80.69$_{\pm0.69}$ | 54.52$_{\pm0.84}$ |
| IsoMax | 87.76$_{\pm0.08}$ | 78.39$_{\pm0.50}$ | 66.67$_{\pm0.36}$ | 76.38$_{\pm0.07}$ | 38.72$_{\pm0.09}$ | 50.06$_{\pm0.44}$ | 77.93$_{\pm0.89}$ | 83.85$_{\pm0.71}$ | 82.26$_{\pm0.56}$ |
| GOOD | 87.62$_{\pm0.28}$ | 77.11$_{\pm0.36}$ | 92.47$_{\pm0.53}$ | 76.03$_{\pm0.70}$ | 50.44$_{\pm0.71}$ | 48.69$_{\pm0.62}$ | 78.74$_{\pm0.83}$ | 78.92$_{\pm0.55}$ | 57.97$_{\pm0.17}$ |
| gDoc | 30.37$_{\pm0.27}$ | 74.54$_{\pm0.41}$ | 94.63$_{\pm0.19}$ | 75.31$_{\pm0.49}$ | 47.55$_{\pm0.40}$ | 39.72$_{\pm0.67}$ | 81.88$_{\pm0.73}$ | 77.17$_{\pm0.47}$ | 74.74$_{\pm0.36}$ |
| GNN_safe | 85.85$_{\pm0.25}$ | 83.26$_{\pm0.31}$ | 90.72$_{\pm0.22}$ | 72.20$_{\pm0.91}$ | 50.66$_{\pm0.51}$ | 43.21$_{\pm0.94}$ | 36.20$_{\pm0.82}$ | 77.67$_{\pm0.37}$ | 82.55$_{\pm0.61}$ |
| OODGAT | 80.00$_{\pm0.97}$ | 77.20$_{\pm0.83}$ | 82.59$_{\pm0.98}$ | 55.80$_{\pm0.87}$ | 39.44$_{\pm0.95}$ | 36.59$_{\pm0.32}$ | 57.89$_{\pm0.92}$ | 64.30$_{\pm0.40}$ | 78.39$_{\pm0.38}$ |
| ARC | 71.02$_{\pm0.20}$ | 72.94$_{\pm0.88}$ | 70.67$_{\pm0.59}$ | 60.22$_{\pm0.23}$ | 37.54$_{\pm0.72}$ | 40.21$_{\pm0.49}$ | 72.12$_{\pm0.30}$ | 65.75$_{\pm0.25}$ | 72.13$_{\pm0.74}$ |
| EDBD | 76.85$_{\pm0.48}$ | 62.87$_{\pm0.58}$ | 81.01$_{\pm0.76}$ | 65.34$_{\pm0.16}$ | 36.94$_{\pm0.40}$ | 35.87$_{\pm0.74}$ | 76.28$_{\pm0.63}$ | 80.05$_{\pm0.79}$ | 70.34$_{\pm0.53}$ |
| **Ours(OGA)** | **90.02$_{\pm0.24}$** | 80.04$_{\pm0.13}$ | **95.97$_{\pm0.59}$** | **78.39$_{\pm0.09}$** | 51.10$_{\pm0.11}$ | 48.79$_{\pm0.74}$ | 83.79$_{\pm0.25}$ | 85.34$_{\pm0.39}$ | 87.53$_{\pm0.18}$ |

objectives: Unknown-Class Rejection (UCR) and Known-Class Node Classification. We compare OGA with several state-of-the-art methods in terms of both rejection and classification performance, evaluating the effectiveness of our approach in handling unlabeled data uncertainty in open-world graph learning scenarios.

We perform comprehensive evaluations on nine datasets, including citation networks, e-commerce networks, and knowledge graphs, to validate the robustness and scalability of OGA. Our results demonstrate that OGA consistently outperforms existing methods, achieving superior Unknown-Class Rejection (UCR) and accuracy for both known and unknown classes. Additionally, OGA significantly enhances the performance of graph neural networks (GNNs) on incomplete graphs, restoring classification accuracy and improving model robustness.

The detailed analysis of our experimental results shows that OGA not only excels in rejecting unknown-class nodes but also ensures high classification accuracy for known-class nodes. Furthermore, we provide a thorough comparison of different backbones, including GCN, GAT, and GraphSAGE, demonstrating the versatility and practical applicability of OGA across various graph learning tasks.

### A.12.1 CLASSIFICATION PERFORMANCE FOR KNOWN CLASSES

Table 10 displays the classification accuracy for known-class nodes across different methods and datasets. OGA consistently outperforms all other methods in known-class classification, achieving the highest accuracy on Cora (90.02%) and Pubmed (95.97%). These results demonstrate that OGA excels in classifying known-class nodes, which is critical for real-world applications where high classification accuracy is required. Moreover, OGA shows strong performance on datasets with more complex class structures, such as WikiCS and History, where it ranks first or second in accuracy. This highlights the robustness and versatility of OGA in a variety of graph-based tasks.

### A.12.2 UNKNOWN REJECTION PERFORMANCE

Table 11 presents the performance of various methods in terms of Unknown-Class Rejection (UCR) and Unknown-Class Accuracy (UCAcc) across different datasets. The performance is ranked separately for UCR and UCAcc, with the best-performing methods highlighted in red, second-best in blue, and third-best in orange.

Our proposed method, OGA, demonstrates outstanding performance, particularly in datasets such as Cora, Pubmed, and WikiCS. OGA achieves the highest UCR of 94.2% on Cora and 82.3% UCAcc, which significantly outperforms other methods like ORAL, OpenWGL, and IsoMax. Notably, OGA performs well even in challenging datasets like Children and arXiv, where other methods show weaknesses in either UCR or UCAcc. This indicates that OGA not only excels in rejecting unknown-class nodes but also ensures high accuracy in identifying these nodes, making it a highly effective solution for open-world graph learning tasks.

**Table 11:** *UCR performance in Aspect 2 (Full). Each item is expressed as **Coverage/Precision**(%).*

| Dataset | Cora | Citeseer | Pubmed | arXiv | Children | Ratings | History | Photo | Wikics |
|---|---|---|---|---|---|---|---|---|---|
| ORAL | 85.2 / 33.6 | 74.7 / 30.3 | 75.0 / 39.1 | 73.3 / 31.4 | 54.0 / 12.7 | 80.1 / 7.4 | 45.8 / 8.1 | 21.5 / 37.2 | 32.8 / 7.2 |
| OpenWgl | 37.9 / 91.3 | 39.2 / 83.6 | 58.5 / 97.1 | 64.1 / 72.4 | 37.8 / 12.3 | 63.2 / 18.7 | 17.4 / 24.9 | 82.6 / 69.0 | 59.4 / 18.6 |
| OpenIMA | 80.5 / 64.0 | 62.9 / 60.2 | 85.3 / 56.4 | OOM | OOM | 72.7 / 31.3 | 50.2 / 25.5 | 64.5 / 54.6 | 69.1 / 33.9 |
| OpenNCD | 27.7 / 87.1 | 59.3 /73.8 | 54.1 / 94.7 | OOM | OOM | 39.1 / 10.9 | 22.8 / 11.0 | 67.4 / 19.5 | 48.5 / 19.4 |
| IsoMax | 67.3 / 65.7 | 41.4 / 78.8 | 70.8 / 87.9 | 59.2 / 52.4 | 50.1 / 13.0 | 54.9 / 10.2 | 72.3 / 24.7 | 58.7 / 47.3 | 38.6 / 13.5 |
| GOOD | 72.8 / 71.5 | 76.1 / 76.5 | 78.2 / 76.4 | 88.0 / 51.8 | 81.4 / 12.9 | 77.6 / 31.9 | 89.7 / 20.2 | 86.5 / 40.5 | 78.3 / 17.3 |
| gDoc | 71.9 / 71.2 | 77.4 / 43.6 | 79.8 / 57.9 | 80.6 / 64.3 | 75.2 / 9.3 | 75.3 / 10.5 | 74.7 / 22.1 | 87.9 / 48.4 | 79.2 / 59.6 |
| GNN_Safe | 90.0 / 43.1 | 92.3 / 38.2 | 91.5 / 40.8 | 85.4 / 40.0 | 88.6 / 14.0 | 88.0 / 14.4 | 72.5 / 36.3 | 95.8 / 41.7 | 87.7 / 30.2 |
| OODGAT | 89.5 / 51.6 | 86.9 / 29.7 | 55.6 / 68.8 | 96.1 / 22.7 | 86.7 / 21.2 | 97.3 / 13.5 | 92.9 / 10.4 | 79.3 / 53.5 | 83.4 / 17.8 |
| ARC | 83.4 / 39.5 | 72.5 / 61.6 | 72.0 / 45.2 | 65.3 / 33.6 | 65.7 /11.8 | 73.2 / 22.6 | 75.4 / 13.9 | 67.2 / 21.5 | 43.1 / 11.7 |
| EDBD | 54.3 / 72.0 | 51.8 / 26.1 | 68.9 / 30.7 | 62.2 / 22.6 | 39.5 / 16.8 | 41.4 / 12.3 | 52.7 / 43.3 | 52.4 / 27.6 | 39.7 / 16.5 |
| **Ours (OGA)** | 94.2 / 82.3 | 93.9 / 65.1 | 79.6 / 98.0 | 91.7 / 60.5 | 96.9 / 24.3 | 81.8 / 33.7 | 82.5 / 37.1 | 89.6 / 72.4 | 96.4 / 48.9 |

### A.12.3 SEMANTIC SIMILARITY ANALYSIS

We analyze the semantic relationships between different node types to better understand the structure of the annotated graphs. Table 12 presents the average semantic similarity across four node pair categories: known-to-known, known-to-generate, generate-to-generate, and generate-to-unknown.

As shown, the known-to-known similarity scores are consistently high across all datasets, typically exceeding 70%. This indicates that nodes belonging to established categories form tight semantic clusters, which facilitates reliable supervision and structured representation learning. In contrast, the known-to-generate similarity scores are significantly lower, often around 31%–38%. The pronounced gap suggests that generated nodes introduce substantial semantic divergence from known categories, thereby enriching the graph with novel concepts rather than merely replicating existing semantics.

The generate-to-generate similarity values lie between the two extremes, averaging around 43%–50%. This moderate intra-group similarity reflects a controlled diversity among generated nodes: while they maintain some degree of semantic coherence, they avoid excessive redundancy. Such diversity is beneficial for covering a broader semantic space without collapsing into trivial duplications.

Additionally, the generate-to-unknown similarity scores are relatively high, generally exceeding 58%. This observation suggests that generated nodes successfully capture latent semantic attributes aligned with unknown classes, potentially serving as intermediaries that bridge the gap between labeled and unlabeled semantic spaces. The ability of generated nodes to maintain proximity to unknown nodes while preserving internal diversity supports the effectiveness of the annotation framework in open-world environments.

Collectively, these results demonstrate that the generated labels not only diversify the semantic landscape beyond known categories but also establish meaningful connections to unknown nodes, thereby facilitating both annotation quality and downstream classification performance.

**Table 12:** *Semantic similarity between different datasets (values are expressed as %)*

| Similarity | Cora | Citeseer | Pubmed | arXiv | Children | Ratings | History | Photo | Wikics |
|---|---|---|---|---|---|---|---|---|---|
| **known to known** | 75.01 | 74.03 | 75.07 | 80.02 | 71.04 | 69.08 | 74.06 | 76.04 | 73.09 |
| **known to generate** | 35.12 | 36.37 | 34.11 | 37.34 | 32.10 | 31.25 | 35.09 | 38.21 | 33.17 |
| **generate to generate** | 50.13 | 48.04 | 46.17 | 50.22 | 45.18 | 43.16 | 46.31 | 49.17 | 44.13 |
| **generate to unknown** | 60.14 | 59.02 | 58.17 | 63.09 | 57.12 | 54.14 | 59.03 | 62.24 | 58.06 |

### A.12.4 PERFORMANCE OF GLA

**Node Classification Accuracy Across Different Graph Inputs.** In Table 13, we present the node classification accuracy for different graph inputs, including the original graph, incomplete graph, and graph completed by our method, OGA. The results show that OGA significantly improves the classification accuracy compared to incomplete graphs, especially on datasets like Pubmed, where GCN's accuracy increases from 47.12% (incomplete) to 87.15% (OGA-completed). Similar improvements are observed for other datasets and GNN architectures, including GAT and SAGE. These findings underline OGA's ability to effectively restore the graph structure, enhancing the performance of GNNs even on incomplete graphs. This is particularly valuable in real-world

**Table 13:** *UCA performance in Aspect 4 (Full).*

| Dataset\Method | Cora | Citeseer | Pubmed | arXiv | Children | Ratings | History | Photo | WikiCS |
|---|---|---|---|---|---|---|---|---|---|
| GCN (Lower) | 61.62 | 58.67 | 47.12 | 55.91 | 42.65 | 37.52 | 75.48 | 71.76 | 60.40 |
| GCN (Ours) | 78.15 | 65.72 | 87.15 | 63.24 | 47.61 | 40.15 | 77.21 | 73.41 | 71.05 |
| GCN (Upper) | 85.56 | 72.71 | 87.49 | 70.14 | 45.60 | 38.87 | 80.95 | 76.47 | 80.76 |
| GAT (Lower) | 60.52 | 58.51 | 46.33 | 57.32 | 46.39 | 37.22 | 76.32 | 73.42 | 60.86 |
| GAT (Ours) | 79.26 | 69.50 | 86.45 | 65.31 | 47.21 | 41.23 | 78.35 | 75.68 | 73.12 |
| GAT (Upper) | 87.04 | 74.08 | 86.19 | 72.65 | 49.40 | 39.12 | 82.10 | 79.70 | 84.04 |
| SAGE (Lower) | 60.29 | 58.55 | 47.56 | 56.48 | 42.84 | 37.85 | 75.87 | 72.42 | 61.41 |
| SAGE (Ours) | 79.11 | 66.04 | 87.19 | 62.76 | 45.12 | 41.02 | 78.02 | 76.21 | 72.24 |
| SAGE (Upper) | 86.67 | 73.37 | 87.80 | 71.59 | 46.67 | 39.01 | 81.63 | 76.37 | 83.06 |

scenarios where graph data is often incomplete or missing, and OGA's graph completion capability ensures robust performance despite such challenges.

**Convergence Analysis.** We analyze the convergence behavior of three backbone models—GCN, GAT, and GraphSAGE—on the **Children** dataset under three graph supervision regimes: lower bound (incomplete graph), upper bound (oracle graph), and our proposed OGA-enhanced graph. As illustrated in Fig 5 and Fig 6 , across all architectures, the OGA-based variants consistently exhibit faster initial loss decline and smoother test accuracy trajectories compared to the lower-bound baseline, indicating that our framework enables more efficient optimization and more stable convergence.

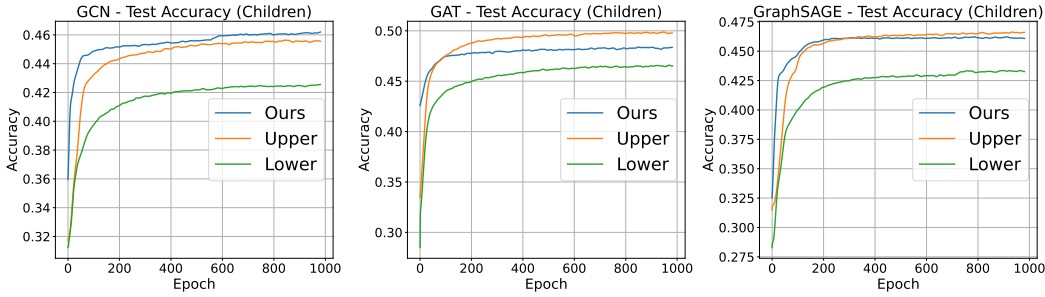

**Figure 5:** *Test accuracy convergence curves of GCN, GAT, and GraphSAGE on the **Children** dataset under three conditions: lower bound, upper bound, and OGA-enhanced graph. The curves represent the model's performance across epochs.*

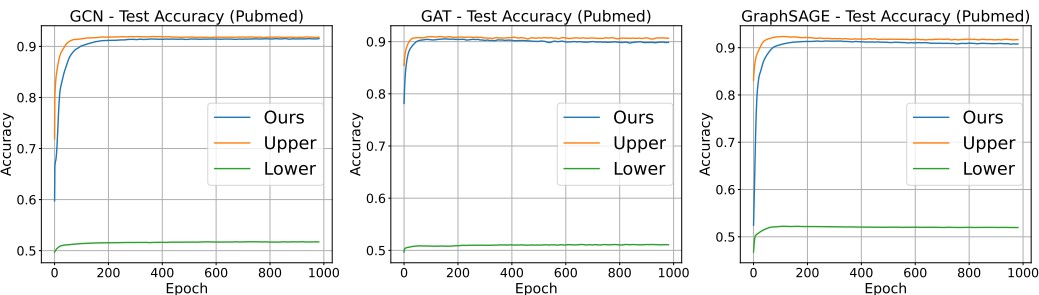

**Figure 6:** *Test accuracy convergence curves of GCN, GAT, and GraphSAGE on the **Pubmed** dataset under three conditions: lower bound, upper bound, and OGA-enhanced graph. The curves represent the model's performance across epochs.*

The Children dataset is characterized by ambiguous class semantics and overlapping community structures, which pose significant challenges for structure-dependent models. In this context, GCN and GraphSAGE particularly benefit from the prior knowledge introduced by OGA. The pre-initialized graph encoder and the ALT module effectively encode high-level ontological priors, allowing the

model to start from a semantically meaningful graph topology, thereby improving gradient stability and accelerating early-stage learning.

As training progresses, the test accuracy curves of the OGA variants consistently improve, achieving competitive or near-optimal performance. The OGA-based models maintain smooth convergence with significantly lower oscillation amplitudes across epochs, compared to the upper-bound setting. This smooth convergence reflects the regularization effects introduced by the topology-aware smoothness loss and the semantic separation margin, which help guide the model toward globally consistent representations while maintaining robustness against label noise. The Children dataset, with its fine-grained class boundaries, exhibits considerable label noise, which the OGA framework effectively mitigates.

The smooth convergence of OGA-based models indicates the role of the GLA module, which continuously refines label assignments by leveraging multi-granularity structural signals. By aligning node predictions with their community-level semantics and mitigating inconsistencies across communities, GLA ensures a steady flow of informative gradients, even in the absence of complete supervision, thereby enhancing the model's robustness and accuracy throughout the training process.

Similarly, the Pubmed dataset, which is characterized by a relatively clean and well-defined class structure, demonstrates robust performance across all GNN backbones. OGA-based models again show smooth convergence in the test accuracy curves, outperforming the upper bound model in several cases. The semantic regularization provided by OGA helps the model avoid overfitting, even with the presence of noisy or ambiguous data, ensuring that the model generalizes well. The GLA module continues to refine label assignments, ensuring that the node-level predictions align well with community-level semantics. This results in a steady and reliable improvement in accuracy, with fewer fluctuations than the baseline models, particularly in scenarios where class boundaries are less well-defined.

### A.13 INTERPRETABILITY ANALYSIS

#### A.13.1 ABLATION STUDY ON ALT

In this ablation study, we aim to investigate the contribution of ontology-based concept modeling within the ALT framework by comparing it with a variant that removes the concept modeling step. This variant is denoted as *w/o Concept Modeling*, where the key component—the ontology-based prototype construction and its corresponding distance-based classification in the ontology space—are entirely omitted. In its place, final predictions are generated using a conventional softmax layer that is directly applied to the enhanced node representations without any further semantic abstraction or alignment with the ontology. This modification effectively transforms the ALT model into a traditional classifier, stripping away the semantic depth and the context-driven confidence-aware rejection mechanism outlined in Equation 3.

**Table 14:** *ALT ablation study across nine datasets.*

| Method | Cora | Citeseer | Pubmed | arXiv | Children | Ratings | History | Photo | WikiCS |
|--------|------|----------|--------|-------|----------|---------|---------|-------|--------|
| w/o CM | 41.25 / 62.31 | 39.24 / 19.12 | 52.10 / 21.75 | 48.91 / 15.20 | 28.63 / 10.94 | 30.41 / 9.53 | 29.32 / 4.12 | 41.29 / 34.90 | 43.12 / 19.32 |
| Ours | 94.87 / 82.66 | 93.00 / 65.19 | 79.76 / 98.53 | 91.91 / 50.13 | 96.98 / 28.34 | 81.75 / 33.62 | 96.57 / 24.78 | 82.24 / 37.56 | 89.07 / 68.09 |

As illustrated in Table 14, removing the ontology-based concept modeling leads to consistent performance degradation across all benchmark datasets. Specifically, on *Cora*, we observe a substantial drop in both the unknown-class rejection rate and the rejection accuracy, with values decreasing from 0.9487 to 0.5489 and 0.8266 to 0.7246, respectively. These metrics reflect the model's diminished capacity to accurately detect out-of-distribution (OOD) nodes, highlighting the importance of structured concept representations in ensuring reliable OOD detection. The impact is more pronounced on datasets with high structural noise or semantically ambiguous boundaries, such as *Ratings* and *History*. In these cases, the unknown-class accuracy falls dramatically from 0.3362 and 0.2478 to 0.0953 and 0.0412, respectively. These results underscore a critical limitation of relying solely on a softmax-based classifier: when confronted with complex or poorly-supervised environments, the model struggles to generalize to unseen or emerging categories. This leads to significant misclassifications, which further highlights the shortcomings of a flat classification approach in open-world settings.

These findings conclusively demonstrate that the ontology-aware prototype modeling component is not merely an auxiliary feature, but a vital element of the ALT framework. By incorporating class-specific semantic anchors and promoting geometric separability in the representation space, prototypes establish a robust foundation for OOD node rejection and the construction of interpretable decision boundaries. Without this mechanism, the model loses its inductive bias, which is crucial for open-world generalization, particularly when faced with scenarios where labeled data is scarce or structural signals are weak. Consequently, this study reinforces the importance of integrating ontology-based concept modeling for achieving accurate, scalable, and interpretable predictions in open-world graph learning tasks.

### A.13.2 ENTROPY-BASED ANALYSIS OF SOFTMAX DISTRIBUTIONS.

To quantitatively examine the behavior of out-of-distribution (OOD) nodes under our model, we analyze the softmax output distributions of both in-distribution (ID) and OOD nodes in the test set. Specifically, we compute the Shannon entropy for each node's softmax probability vector (restricted to known classes) as a measure of prediction confidence and distribution sharpness. Formally, for a node $i$, the entropy is defined as $\mathcal{H}_i = -\sum_{c=1}^{C} p_{ic} \log p_{ic}$, where $p_{ic}$ denotes the predicted probability for class $c$.

As visualized in Figure 8, a stark contrast emerges between the two groups: ID nodes exhibit a highly concentrated entropy distribution near zero, with the majority of samples falling below $\mathcal{H} < 0.2$. This indicates that the softmax outputs of ID nodes are sharply peaked—suggesting confident predictions dominated by a single class. In contrast, OOD nodes display a noticeably broader entropy spectrum, with a substantial portion lying in the range of $0.2 < \mathcal{H} < 1.0$, and a long tail extending even beyond $\mathcal{H} = 1.4$. This evidences that OOD nodes produce significantly smoother and more uncertain softmax distributions.

This entropy gap substantiates our hypothesis: OOD nodes are inherently less confident in classification over the known label space, leading to more uniform probability distributions. Consequently, these statistical patterns provide strong empirical justification for our confidence-based rejection mechanism, wherein nodes with low maximum softmax scores (i.e., high entropy) are rejected as uncertain or potentially OOD.

### A.13.3 ABLATION STUDY ON GLA

To further validate the effectiveness of semantic-aware community detection in GLA, we conduct additional ablation studies on two datasets: **Citeseer** and **Pubmed**. We specifically evaluate the impact of removing semantic similarity during community partitioning (*w/o Semantic Community*), as well as the effect of skipping the community-level annotation distillation step (*w/o Intra-community Distillation*). The results are summarized in Table 6.

**Metric Definitions.** We define *Redundancy* (RE) as the total number of distinct labels generated across all communities. A lower RE indicates better semantic grouping and reduced label duplication. *Consistency* (Con) measures the average pairwise cosine similarity among node-level annotations within the same community, reflecting the semantic coherence of community assignments. A higher Con value indicates greater internal semantic alignment among nodes.

**Impact of Removing Semantic Community Detection.** When semantic similarity is removed from the community detection objective (i.e., setting $\gamma = 0$ in Eq. 5), community partitioning relies solely on graph topology. As shown in Table 6, this results in a substantial degradation across all evaluation metrics. Specifically, the number of distinct labels (RE) nearly doubles, increasing from 9 to 17 on Citeseer and from 5 to 11 on Pubmed. This indicates severe over-segmentation, where nodes that could have been semantically clustered are now treated as separate groups, leading to redundant and fragmented annotations.

Furthermore, intra-community semantic consistency (Con) drops significantly—on Citeseer, from 0.84 to 0.70, and on Pubmed, from 0.86 to 0.73. This suggests that without semantic guidance, nodes grouped into the same community become semantically dissimilar, reducing the coherence of in-context prompts used for LLM-based annotation.

Finally, the downstream classification accuracy also suffers notable declines, with a reduction of 4.8% on both Citeseer and Pubmed. These drops are consistent with the hypothesis that noisier,

semantically inconsistent communities weaken the quality of supervision signals, thereby impairing model training and generalization.

These observations collectively demonstrate that semantic-aware community detection is essential not only for annotation efficiency (by reducing redundant labels) but also for annotation quality (by maintaining high semantic coherence), which ultimately translates into improved downstream performance.

**Impact of Removing Intra-community Distillation.** In a complementary ablation, we disable the community-level label distillation process (i.e., skipping Eq. 7) and instead apply LLM-based annotation directly to each node without selecting representative nodes or aggregating shared labels. Without distillation, each node independently queries the LLM, leading to fragmented annotations within the same community.

This ablation setting is expected to harm both annotation consistency and computational efficiency. Without intra-community distillation, nodes within the same structural community are annotated independently, resulting in increased semantic noise and lower internal coherence. In practice, this results in reduced Con scores, more fragmented label distributions, and lower annotation reliability for model learning. Moreover, the annotation cost grows substantially, as the number of LLM queries reverts closer to a naïve node-by-node setting, undermining one of GLA's major efficiency advantages.

**Conclusion.** The results of these ablation studies reinforce that both semantic-aware community detection and intra-community distillation are critical components of GLA. Removing either component leads to consistent degradation in annotation redundancy, consistency, and downstream classification performance, validating their importance in achieving scalable, high-quality open-world graph annotation.

### A.13.4 INTERPRETABILITY ANALYSIS OF GLA: A CASE STUDY

To illustrate the interpretability and effectiveness of the proposed Graph Label Annotator (GLA), we conduct a detailed case study on the *Cora* dataset. Specifically, we focus on the subset of nodes that have been rejected as unknown-class instances based on a prior open-set classification stage. These nodes are considered out-of-distribution (OOD) samples, lacking ground-truth annotations, and representing real-world cases where new or emerging categories are not covered by the training data.

This open-world setting poses a significant challenge, as the model must infer coherent and generalizable semantic labels from both textual content and structural context without access to labeled supervision. GLA is specifically designed to address this challenge by combining graph community structures with large language model (LLM) prompting to produce compact, consistent, and human-interpretable labels for such unknown-class nodes.

Table 15: *GLA-based Community Merging Results.*

| Merged Label | Community IDs | # Nodes | Merged Semantics |
|:---:|:---:|:---:|:---:|
| **Rule_Learning** | 2, 3 | 47 | Rule-based decision strategies |
| **Genetic_Algorithms** | 5, 11 | 87 | Evolutionary optimization |
| **Bayesian_Cluster_Detection** | 9, 28 | 91 | Bayesian inference + clustering |
| **Probabilistic_Classification** | 10 | 4 | Statistical learning |
| **Neural_Networks** | 6, 21, 36 | 60 | Deep learning architecture |
| **Structural_Equation_Models** | 12, 14, 16 | 55 | Latent variable modeling |
| ... | ... | ... | ... |

To provide a clearer view of how GLA organizes unlabeled nodes into coherent semantic clusters, we summarize representative community merging results in Table 15. Each row lists a merged label along with the communities it consolidates, the total number of nodes involved, and the overarching semantic theme. This summary illustrates GLA's ability to aggregate semantically consistent subgraphs and abstract meaningful labels. For instance, **Rule_Learning** results from merging Communities 2 and 3, which share strong connections to decision procedures and symbolic reasoning. Similarly, **Genetic_Algorithms** unifies Communities 5 and 11, both centered around adaptive, evolutionary

mechanisms. Even small communities like Community 10—consisting of only 4 nodes—are assigned a well-aligned label (**Probabilistic_Classification**), showcasing GLA's robustness in low-resource cases.

The case study begins with the application of Louvain-based community detection on the filtered subgraph containing only out-of-distribution (OOD) nodes. This process yields over 30 structurally coherent communities of varying sizes. Among them, Community 2 and Community 5 are selected for close analysis. Community 2 contains 36 nodes whose textual contents revolve around decision-making strategies, rule-based classification, and symbolic control. Community 5 includes more than 60 nodes and focuses on evolutionary adaptation, optimization under uncertainty, and population-based methods. In contrast, smaller communities such as Community 10 (with only 4 nodes) and Community 8 (also 4 nodes) capture fine-grained semantics, often forming local conceptual clusters, for instance on probabilistic modeling or visual mapping.

GLA performs annotation in two stages: intra-community distillation and inter-community fusion. The entire annotation pipeline is closely aligned with the theoretical formulation introduced in Section 3.

In the first stage, Louvain-based clustering is conducted over the OOD subgraph using a modularity objective augmented with semantic similarity, as defined in Equation 5. This formulation integrates both graph structure and text embedding similarity to obtain communities that are topologically and semantically coherent. Within each community, nodes are sorted by degree and split into high- and low-degree groups based on the median. Following Equation 6, low-degree nodes are assigned priority for direct annotation via LLM, while high-degree nodes derive their annotations from the surrounding labeled neighborhood, minimizing the number of LLM calls required.

Once node-level annotations are obtained, a representative set of nodes is selected using a combination of topological centrality and semantic diversity. Their annotations are aggregated via a distillation process to produce a single community-level label, as described in Equation 7. For example, Community 2 is distilled into the label **Rule_Learning**, while Community 5 is assigned **Genetic_Algorithms**, both accurately reflecting their internal themes.

In the second stage, GLA conducts inter-community label fusion to merge overlapping or semantically close communities. Pairwise similarity between community-level annotations is computed based on cosine similarity of their embedding representations, and communities exceeding a threshold are merged iteratively. The resulting merged label is generated via LLM-based semantic fusion, as formalized in Equation 8. For instance, Community 9 and Community 28, both linked to Bayesian modeling, are merged and relabeled as **Bayesian_Cluster_Detection**, ensuring semantic compactness and label reuse.

Notably, even small communities like Community 10 (only 4 nodes) are handled robustly by GLA through direct prompting and high-confidence annotation, demonstrating adaptability across varying community sizes.

Following intra-community labeling, GLA performs inter-community label fusion. This stage aims to reduce label redundancy and improve generality by merging communities with high semantic similarity. To achieve this, GLA computes pairwise cosine similarity between the embedding vectors of each community's distilled annotation, obtained via an embedding model. If the similarity exceeds a dynamic threshold, communities are iteratively merged, and a new, more abstract label is generated using a fusion-based prompt strategy. For example, Community 9 and Community 28 are merged due to their shared focus on Bayesian reasoning and cluster inference. Their individual annotations are refined into a unified label: **Bayesian_Cluster_Detection**. This fused label is then assigned to all nodes in the merged cluster, ensuring consistency and abstraction.

Through this two-stage process, GLA outputs a compact and semantically interpretable label space. In this case, the final set of generated labels includes: **Rule_Learning**, **Neural_Networks**, **Case_Based**, **Genetic_Algorithms**, **Uncertainty_Modeling**, **Risk_Learning**, **Probabilistic_Classification**, **Bayesian_Cluster_Detection**, **Instance_based_Learning**, **Structural_Equation_Models**, **Memory_Based_Learning**, and **Recurrent_Neural_Networks**. These labels are carefully curated to be concise (no more than three words), compositionally meaningful, and aligned with well-established subfields in machine learning and statistical modeling.

From an interpretability standpoint, GLA's advantage lies in its principled integration of topological signals and semantic reasoning. By leveraging homophily for node selection, modularity for structure detection, and semantic similarity for label merging, GLA constructs a layered annotation pipeline that is both transparent and scalable. Furthermore, the use of degree-aware prompting reduces unnecessary LLM invocations, lowering computational cost while preserving annotation quality.

In summary, this case study highlights the strength of GLA in distilling high-quality semantic prototypes from noisy unlabeled graphs. Its design enables both fine-grained labeling within communities and abstraction across communities, offering a promising solution for scalable and interpretable annotation in open-world graph learning.

## A.14 ROBUSTNESS ANALYSIS

### A.14.1 HYPERPARAMETER ANALYSIS OF $\lambda$

To evaluate the role of the sharpness parameter $\lambda$, which controls the slope of the distance-based softmax function in Eq. 3, we conduct controlled experiments across multiple datasets with $\lambda \in \{0.1, 0.3, 0.5, 0.7, 0.9\}$. The results are visualized in Fig. 7, highlighting key performance metrics: known-class accuracy, rejection coverage, and rejection precision.

As $\lambda$ increases, both classification accuracy on known classes and rejection coverage improve. Specifically, classification accuracy peaks at $\lambda = 0.5$, with significant improvements across all datasets. For example, on the Cora dataset, known-class accuracy reaches 0.891 at $\lambda = 0.5$, while the rejection rate increases to 0.910. This trend is observed in other datasets as well, with the rejection rate steadily rising as $\lambda$ increases, indicating a stronger separation in the ontology space and a better capacity for OOD rejection.

However, as $\lambda$ increases beyond a certain threshold, performance improvement diminishes, and in some cases, it leads to undesirable effects. In particular, the rejection accuracy begins to drop as $\lambda$ becomes too large. For instance, when $\lambda$ increases to 1.0, rejection accuracy decreases on several datasets, such as Cora and Wikics, due to overconfident misclassification of in-distribution nodes as OOD. This over-sharpening causes the decision boundaries to become too rigid, misclassifying highly uncertain nodes.

The results suggest that $\lambda$ controls a Pareto frontier between classification precision and rejection reliability. An optimal range for $\lambda$ is observed to be between 0.5 and 0.8, where the balance between confident classification of known categories and accurate rejection of OOD samples is most effectively maintained. At lower values of $\lambda$, the decision boundary remains soft, resulting in a lower rejection rate but higher rejection accuracy. However, the overall classification accuracy is suboptimal, as insufficient inter-class separation in the ontology space limits the model's ability to distinguish between different classes.

In conclusion, these experiments support the hypothesis that $\lambda$ plays a crucial role in regulating the trade-off between the accuracy of classification and the reliability of rejection. Moderate values of $\lambda$ provide a balanced trade-off, while higher values lead to overfitting and overconfidence in predictions, thereby reducing rejection accuracy. The optimal range of $\lambda \in [0.5, 0.8]$ ensures both strong classification performance and robust rejection behavior, aligning with the open-world assumption of maintaining uncertainty for unknown categories while classifying known categories with confidence.

### A.14.2 EFFECT OF UNKNOWN CLASS RATIO ON ALT.

To evaluate the robustness of our model under varying open-world conditions, we conduct experiments on the WikiCS dataset by progressively increasing the ratio of unknown classes from 0.1 to 0.8. WikiCS is selected for this study due to its relatively large number of categories and homophilic graph structure, which ensures sufficient semantic granularity and stable message passing. These properties make it particularly suitable for simulating different degrees of open-worldness and for assessing the model's capacity to balance known-class classification and unknown-class rejection.

As illustrated in Figure 9, the model maintains strong performance in both OOD rejection and unknown-class classification accuracy across a wide range of unknown class ratios. Notably, as the unknown ratio increases from 0.1 to 0.3, OOD accuracy rises from 27.74% to 59.85%, accompanied

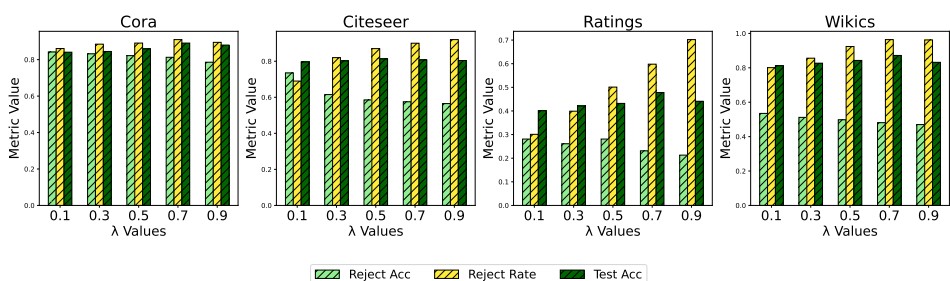

**Figure 7:** *Effect of λ on key performance metrics. As λ increases, classification accuracy on known classes improves steadily until it reaches a saturation point, while the rejection rate also increases, indicating stronger separation in the ontology space. However, when λ becomes too large, leading to overly sharp boundaries, rejection accuracy starts to decrease due to overconfident misclassification of out-of-distribution (OOD) samples. An optimal trade-off is observed in the range of $\lambda = 0.5 \sim 0.8$.*

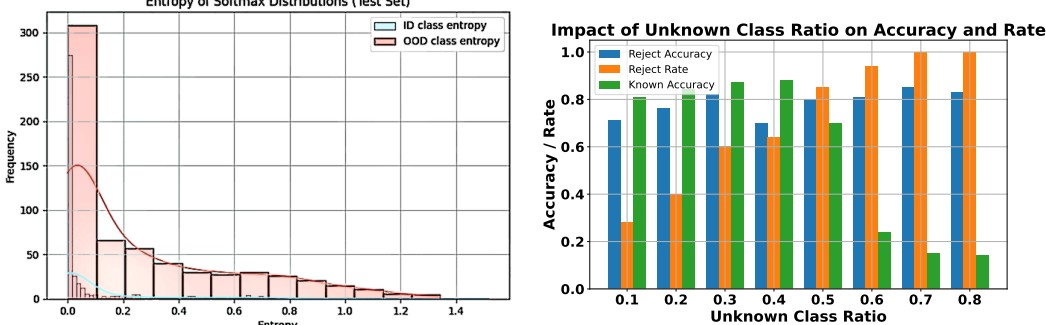

**Figure 8:** *Shannon Entropy Analysis for OOD and ID Nodes.*

**Figure 9:** *Unknown-Class Rejection and Accuracy on the WikiCS Dataset.*

by an increase in rejection rate from 71.14% to 81.63%. This trend suggests that the presence of more distinguishable OOD samples facilitates the model's ability to learn a clearer separation between known and unknown instances.

Interestingly, the rejection rate does not increase monotonically with the unknown class ratio. A slight dip is observed at ratio 0.4 (69.85%), before the rejection rate recovers and stabilizes in the 80–85% range at higher ratios. This intermediate decline may reflect a transitional phase where the inter-class boundaries between known and unknown nodes become less distinct due to more balanced class distributions, thereby increasing the difficulty of rejection.

In terms of OOD accuracy, the model exhibits a near-linear improvement up to a ratio of 0.6, reaching 93.56%, and achieves perfect separation (100%) when the unknown ratio exceeds 0.7. These results highlight the model's capacity to generalize to unseen categories under increasingly open-world scenarios. However, this improvement is accompanied by a decline in in-distribution (ID) performance. Known-class accuracy remains relatively stable up to ratio 0.5 (89.11%) but experiences a sharp degradation to 24.35% at ratio 0.6, and further declines to approximately 15% when the majority of classes are unknown. This sharp drop indicates a critical threshold beyond which the reduced representation of known classes substantially impairs the model's ability to preserve ID classification performance.

Overall, this experiment underscores a fundamental trade-off: as the proportion of unknown classes increases, the model becomes more proficient at identifying and rejecting OOD nodes, but this comes at the cost of degrading ID classification accuracy. The turning point appears to occur between ratios 0.5 and 0.6, marking a regime shift in model behavior. For practical deployment, this emphasizes the importance of calibrating OOD-aware systems to maintain a balanced performance between safety (via rejection) and reliability (via ID classification).

### A.14.3   HYPERPARAMETER ANALYSIS OF $\alpha$ AND $\beta$

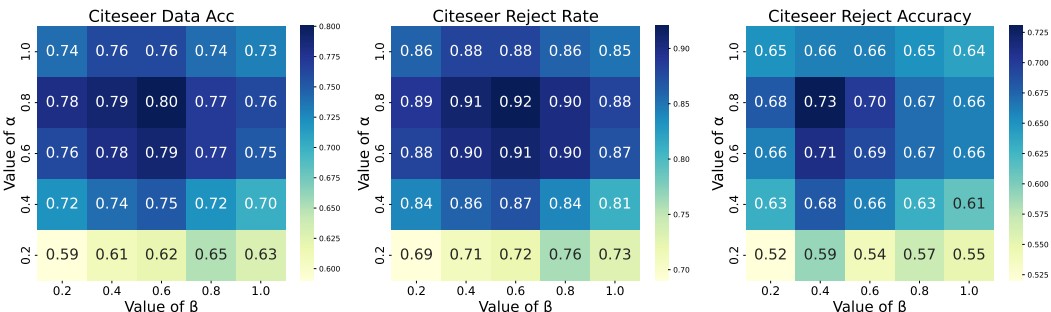

**Figure 10:** *Heatmap showing the impact of $\alpha$ and $\beta$ on Accuracy, UCR, and Unknown Accuracy. The optimal performance is observed at $\alpha = 0.8$ and $\beta = 0.6$, achieving the best balance between known-class accuracy and unknown-class rejection.*

In this section, we analyze the impact of the hyperparameters $\alpha$ and $\beta$ on key performance metrics: Accuracy, Unknown Class Rejection Rate (UCR), and Unknown Class Accuracy. These hyperparameters control the balance between topology regularization and class separation within the model's loss function, influencing the model's ability to classify known-class nodes, reject unknown-class nodes, and correctly identify unknown-class nodes.

The results from Figure 10 indicate that $\alpha$ has a significant effect on Accuracy. When $\alpha = 0.4$, the model achieves its highest known-class accuracy of 76%. However, as $\alpha$ increases, the accuracy starts to decline due to over-smoothing of graph embeddings, which reduces the model's ability to discriminate fine-grained features. This suggests that a moderate value of $\alpha$ is optimal for maintaining classification accuracy, with larger values leading to diminishing returns in terms of performance.

On the other hand, $\beta$ plays a more dominant role in controlling the Unknown Class Rejection Rate (UCR). As $\beta$ increases, the model's ability to reject unknown-class nodes improves, with the best rejection performance occurring at $\beta = 0.6$, where UCR reaches 92%. This result demonstrates that a stronger class separation, governed by $\beta$, enhances the model's capacity to distinguish between

known and unknown classes, thereby improving rejection accuracy. However, as $\beta$ increases beyond this point, the model becomes stricter in its rejection criteria, which negatively affects the model's ability to correctly identify unknown-class nodes.

Indeed, there is a trade-off between UCR and Unknown Accuracy. As $\beta$ increases, the model's rejection criteria for unknown-class nodes become more stringent, causing a decrease in Unknown Accuracy. The highest Unknown Accuracy of 65.1% is observed at $\beta = 0.6$, which, although lower than some baseline methods, is accompanied by superior UCR performance. This indicates that a strong rejection performance often comes at the cost of reduced ability to correctly identify unknown-class nodes.

In conclusion, the optimal hyperparameter configuration for the Citeseer dataset is $\alpha = 0.8$ and $\beta = 0.6$, which strikes the best balance between Accuracy and UCR. This combination is particularly suitable for tasks that prioritize the rejection of unknown-class nodes, offering the best trade-off between rejection performance and classification accuracy. While increasing $\beta$ beyond 0.6 would enhance UCR, it would do so at the expense of Unknown Accuracy. On the other hand, optimizing $\alpha$ closer to 0.5 may offer a better trade-off when the primary goal is maximizing known-class accuracy, as it would help maintain a stronger differentiation between nodes in the graph.

### A.14.4 HYPERPARAMETER ANALYSIS OF SEMANTIC-TOPOLOGY BALANCE $\gamma$

The parameter $\gamma$ controls the relative importance between topological structure and textual semantics during community detection in our GLA module. By interpolating between the structure-only ($\gamma \to 0$) and semantics-only ($\gamma \to 1$) regimes, it determines how well the constructed communities preserve topological integrity while maintaining semantic coherence. This balance directly affects both the quality of the community structure and the cost-efficiency of the LLM-based label annotation.

We report three evaluation metrics across varying values of $\gamma$ on four datasets: *(i)* modularity (structure consistency), *(ii)* semantic consistency (measured as average pairwise cosine similarity of node embeddings within communities), and *(iii)* the number of LLM calls required for annotation. The results are visualized in Fig. 11, where $\gamma \in \{0.0, 0.2, 0.4, 0.6, 0.8, 1.0\}$.

When $\gamma$ is small (e.g., $0.0 \sim 0.2$), the community detection is dominated by graph topology, which tends to group tightly connected nodes regardless of their semantic meaning. As a result, the modularity score is relatively high, reflecting strong structural coherence. However, semantic consistency remains low, and the LLM must process more ambiguous or incoherent textual neighborhoods, leading to a higher number of LLM queries.

On the other end, when $\gamma$ is large (e.g., $0.8 \sim 1.0$), the algorithm favors semantic similarity over graph structure, causing communities to become semantically pure but topologically disconnected. This results in decreased modularity and a fragmented community layout. Furthermore, due to smaller average community size and lower label reusability across communities, the LLM call frequency increases again.

The optimal trade-off emerges at $\gamma = 0.6$, where all three metrics are well balanced. At this setting, communities exhibit both structural cohesion and semantic homogeneity, resulting in improved annotation consistency and a minimal number of LLM queries. These findings empirically validate our design of integrating both structural and semantic signals in GLA and further support the theoretical hypothesis that joint modeling of graph and language yields efficient and coherent in-context prompts.

The results from the four datasets are as follows:

- **Citeseer:** Modularity values range from 0.60 to 0.88, with semantic consistency improving from 0.20 to 0.85 as $\gamma$ increases. The number of LLM calls decreases as $\gamma$ approaches 0.6, then increases with higher $\gamma$ values.
- **Cora:** Similar to Citeseer, modularity values range from 0.61 to 0.87, and semantic consistency improves from 0.21 to 0.86. The LLM calls exhibit the same trend of decreasing and then increasing as $\gamma$ changes.
- **Wikics:** Modularity ranges from 0.63 to 0.91, while semantic consistency starts at 0.25 and increases to 0.89. LLM calls also follow the same trend of increasing and decreasing with $\gamma$ changes.

- **Ratings:** The modularity starts from 0.57 and increases to 0.75, with semantic consistency ranging from 0.29 to 0.92. The number of LLM calls shows a similar behavior to the other datasets.

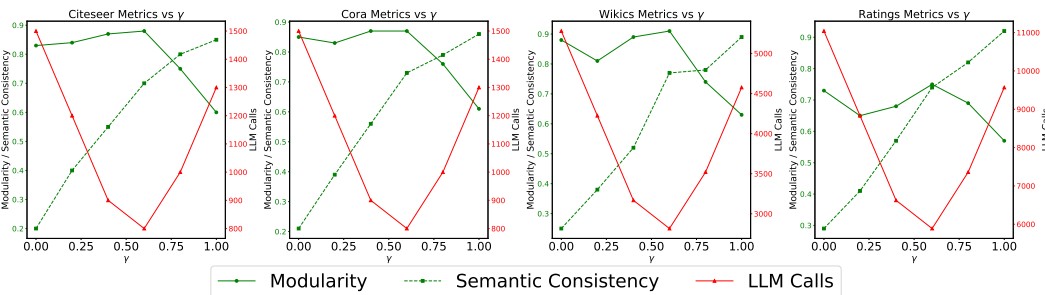

**Figure 11:** *Effect of semantic-topology balance $\gamma$ on modularity, semantic consistency, and LLM calls (Citeseer). A clear optimum is observed at $\gamma = 0.6$.*

### A.15 EFFICIENCY ANALYSIS

#### A.15.1 TIME COMPLEXITY OF ALT

Among the twelve baseline methods included in our evaluation, we selectively report running time for three representative models—**OpenWGL**, **IsoMax**, and **GOOD-D**—to provide a clear and interpretable comparison of computational efficiency. These three methods were chosen to represent a diverse range of methodological categories and complexity levels. Specifically, OpenWGL serves as a representative of full-fledged open-world graph learning frameworks that incorporate latent uncertainty modeling (via variational graph autoencoders) and sophisticated open-set classification mechanisms. Its comprehensive pipeline typically incurs the highest computational cost among all baselines, thus offering an upper bound in our efficiency analysis. IsoMax, on the other hand, represents a lightweight open-set/OOD detection approach based on isotropy maximization loss, which modifies the softmax layer with minimal architectural changes. As such, it provides a lower bound reference for runtime and demonstrates the efficiency of minimal modifications. GOOD-D is a contrastive learning-based OOD detection framework that operates on multiple hierarchical levels (node, graph, and group) and captures intermediate complexity. It offers a balanced perspective on computational cost and detection performance. These three baselines thus form a representative triad that captures the trade-offs between expressiveness, runtime, and detection accuracy. Other methods, such as ORAL, OpenIMA, OODGAT, and OpenNCD, often involve additional components like clustering algorithms, Hungarian matching, or prototype memory structures, which are more sensitive to specific implementations and may introduce variability in timing measurements. Therefore, for reproducibility and conciseness, we restrict the time efficiency analysis to OpenWGL, IsoMax, and GOOD-D, which together provide a meaningful and stable basis for evaluating the efficiency of our proposed method.

**Table 16:** *Running Time (in seconds) of Different Methods on Nine Datasets*

| Dataset\Method | Cora | Citeseer | Pubmed | arXiv | Children | Amazing | History | Photo | WikiCS |
|---|---|---|---|---|---|---|---|---|---|
| OpenWGL | 34.93 | 23.76 | 104.33 | 2223.27 | 727.52 | 91.57 | 232.52 | 1535.24 | 1235.24 |
| IsoMax | 4.21 | 6.01 | 25.58 | 38.12 | 25.33 | 11.86 | 12.92 | 13.20 | 12.79 |
| GOOD | 13.12 | 16.43 | 92.48 | 144.66 | 127.64 | 21.39 | 75.46 | 39.62 | 53.17 |
| **Ours** | **8.91** | **7.30** | **56.84** | **99.23** | **73.83** | **16.84** | **46.44** | **25.43** | **29.48** |

We compare the running time of different methods in Table 16. Our method achieves a favorable balance between accuracy and computational efficiency. For instance, compared to OpenWGL, which incurs extremely high runtime on large-scale graphs such as arXiv (2223.27s) and WikiCS (1235.24s), our model significantly reduces the computational cost to 99.23s and 29.48s, respectively. Furthermore, in comparison with lightweight baselines such as IsoMax and mid-complexity methods

like GOOD, our approach demonstrates competitive efficiency, particularly on medium-scale datasets, while maintaining strong performance in unknown node detection.

The superior efficiency of our approach is primarily attributed to the design of the proposed **ALT module**, whose overall time complexity is given by:

$$\mathcal{O}(knd + |V_l|\bar{d}d + nC_kd + C_k^2d),$$

where $n$ denotes the number of nodes, $d$ the embedding dimension, $k$ the number of propagation steps, $C_k$ the number of class prototypes, $|V_l|$ the number of labeled nodes, and $\bar{d}$ the average node degree. The first term corresponds to the propagation process via $k$-step graph convolution over sparse adjacency matrices, which ensures scalability with graph size. The second term accounts for the construction of concept prototypes through attention-weighted neighborhood aggregation over labeled nodes. The third term captures the cost of classification, which involves computing distances between all nodes and prototype representations, and thus scales linearly with both $n$ and $C_k$. The final term reflects the cost of the multi-term optimization objective, including semantic alignment, smoothness regularization, and separation constraints, which remain manageable in practice.

- **Propagation:** The $k$-step graph convolution over sparse adjacency matrices costs $\mathcal{O}(knd)$, scalable with graph size.
- **Concept Construction:** Attention-weighted neighborhood aggregation is limited to labeled nodes, with cost $\mathcal{O}(|V_l|\bar{d}d)$.
- **Classification:** Prototype-based distance computation scales linearly in $n$ and $C_k$, i.e., $\mathcal{O}(nC_kd)$.
- **Optimization:** Includes semantic, smoothness, and separation terms with cost $\mathcal{O}(|V_l|C_k + nd + C_k^2d)$.

In summary, the overall complexity remains linear in $n$, ensuring the method is efficient and scalable to large graphs. This is reflected in the empirical results, where our approach maintains lower or moderate time consumption across all datasets.

### A.15.2 TIME EFFICIENCY IMPROVEMENT BROUGHT BY GLA

To improve the scalability and efficiency of open-world annotation in text-attributed graphs (TAGs), the proposed **Graph Label Annotator (GLA)** framework significantly reduces reliance on costly node-by-node large language model (LLM) inference. In a naïve baseline setting, each node $v_i$ in the graph is independently passed to an LLM with its associated textual input $T_i$, i.e., $\tilde{y}_i = \text{LLM}(T_i)$. This results in a computational complexity of $\mathcal{O}(n)$ LLM queries, where $n$ is the total number of nodes. Such a method overlooks the structural and semantic homophily inherent in real-world graphs, leading to redundant computation and suboptimal scalability.

GLA addresses this inefficiency by introducing a structure-guided, multi-granularity annotation pipeline. First, it performs a community detection procedure based on a modularity objective that integrates both graph topology $A$ and node-level semantic similarity $\tilde{e}_i^\top \tilde{e}_j$. Formally, the optimization target is defined as:

$$[Q = \frac{1}{2m} \sum_{i,j} \left[ A_{ij} + \gamma \cdot \frac{\tilde{e}_i^\top \tilde{e}_j}{\|\tilde{e}_i\| \cdot \|\tilde{e}_j\|} - (1-\gamma) \cdot \frac{d_i d_j}{2m} \right] \delta(c_i, c_j),] \tag{63}$$

where $\delta(c_i, c_j) = 1$ if nodes $v_i$ and $v_j$ belong to the same community. This formulation allows for the discovery of structurally and semantically cohesive regions of the graph that can be jointly annotated.

Within each community, GLA adopts a degree-aware LLM annotation strategy. Low-degree nodes (i.e., nodes with insufficient local information) are prioritized for LLM querying, while high-degree nodes can inherit annotations from their neighbors via a lightweight allocation mechanism:

$$[\tilde{y}_i = \begin{cases} \text{LLM-Annotation}(T_i, \{T_j \mid v_j \in N(v_i)\}), & \text{if } d_i < \bar{d} \\ \text{Allocation}(T_i, \{T_j, \tilde{y}_j \mid v_j \in N(v_i)\}), & \text{otherwise.} \end{cases}] \tag{64}$$

This significantly reduces LLM inference cost, as only a small fraction of the graph (the low-degree nodes) need direct access to the LLM, while the rest benefit from neighborhood supervision.

Furthermore, GLA performs hierarchical annotation distillation and fusion. For each community $P$, representative nodes are selected based on structural and semantic centrality, and their LLM-derived annotations are aggregated into a unified label $\tilde{y}_P$. To avoid redundancy and promote semantic consistency, similar communities are recursively merged through an LLM-guided fusion process until a target number of cluster-level labels is reached:

$$[\tilde{y}_{P_{ij}}^{\star} = \text{LLM-Fusion}(\tilde{y}_{P_i}, \tilde{y}_{P_j}), \quad \text{Sim}(P_i, P_j) = \frac{1}{|P_i||P_j|} \sum_{m \in P_i} \sum_{n \in P_j} M(\tilde{e}_m, \tilde{e}_n).] \tag{65}$$

Overall, the GLA approach reduces the total number of LLM queries from linear $\mathcal{O}(n)$ to sublinear $\mathcal{O}(n_{\text{low}} + k + \log k)$, where $n_{\text{low}} \ll n$ is the number of low-degree nodes and $k \ll n$ is the number of communities. This not only improves computational efficiency but also enhances label consistency across structurally similar nodes. As a result, GLA provides a scalable, structure-aware solution for open-world graph annotation that aligns with the practical deployment needs of LLM-integrated graph learning systems.

### A.16 DECLARATION OF WRITING ASSISTANCE TECHNOLOGIES

In preparing this manuscript, we used generative artificial intelligence (GenAI) tools—specifically GPT-4o and Grok-4—for language polishing and for assisting with the drafting and revision of ancillary code snippets. These tools were employed solely to improve clarity and readability and to streamline the presentation. Importantly, GenAI was not used for deriving mathematical formulas, designing or implementing key algorithms, or formulating the core scientific insights. All theoretical proofs, algorithmic developments, and experimental validations were carried out independently by the authors to preserve the integrity and originality of the research. We thoroughly reviewed and verified all AI-assisted text to ensure accuracy and consistency with the scientific content, thereby upholding the reliability of the reported results.

