# OpenReview forum: "When LLMs Encounter Open-world Graph Learning: A Fresh View on Unlabeled Data Uncertainty"
_ICLR.cc/2026/Conference — ICLR 2026 Conference Withdrawn Submission_

### Official Review · Reviewer_xUs9 · 2025-10-30

**Soundness:** 3
**Presentation:** 2
**Contribution:** 3
**Rating:** 8
**Confidence:** 3

**Summary:**

This paper proposes OGA, a fully automated LLM-based data annotation framework designed to address unlabeled data uncertainty in open-world graph learning. Specifically, the framework introduces an Adaptive Label Traceability (ALT) module that integrates semantic and topological reasoning for unknown-class rejection (UCR), followed by a LLM-driven Graph Label Annotator (GLA) for unknown-class annotation (UCA). The proposed framework is theoretically grounded, and extensive experiments are conducted from different aspects to demonstrate the effectiveness of the proposed framework.

**Strengths:**

S1. The motivation behind each proposed module is clearly and thoroughly explained.
S2. The proposed integration of semantic and topological optimization within the ontology space is novel.
S3. The authors provide a detailed theoretical analysis of the proposed framework.
S4. The authors conduct comprehensive experiments from various key perspectives to thoroughly validate the effectiveness of the proposed framework.

**Weaknesses:**

W1. Lack of clarity in framework details.
    W1.1. The paper emphasizes that one of its key contributions is leveraging a pre-trained graph-language encoder to obtain node representations. However, no description or implementation details are provided regarding this component. Therefore, it remains unclear whether the “graph-language encoder” refers to (a) an independent, pre-trained module used to initialize node embeddings, or (b) the integrated semantic–topology optimization in the ontology space.
    W1.2. Please include a description of the overall framework illustrated in Figure 1/2 in the figure caption.
W2. The paper emphasizes that leveraging a pre-trained graph-language encoder is a key advantage over graph-only encoders. If the “graph-language encoder” refers to a pre-trained module used to initialize node embeddings, this claim is not empirically supported. Please conduct ablation study on graph-language encoder vs. graph-only/language-only encoders.
W3. The authors claim that the proposed framework addresses the challenge of unlabeled data uncertainty, which is particularly critical in large-scale datasets due to high annotation costs. However, the scalability of the proposed framework is evaluated only on one large-scale graph (i.e., Arxiv), please evalute the scalability of the proposed framework on more large-scale datasets.

**Questions:**

Please see W1.1, W2, and W3.

---

> ### Author Response · Authors · 2025-11-18
>
> >**Q1.The paper emphasizes that one of its key contributions is using a pretrained graph language encoder to obtain node representations. But no description or implementation details of this component are provided. It is thus unclear whether the “graph language encoder” refers to (a) a standalone pretrained module to initialize node embeddings, or (b) an integrated semantic-topological optimization in the ontology space.**
>
> **W1:** We apologize for the lack of clarity in the current draft. In our framework, the “graph language encoder” refers to an integrated module that combines textual semantics with graph topology, rather than a mere standalone initializer. Concretely, we first obtain text embeddings for node TAGs using a pretrained language model, and then feed these embeddings into a graph encoder that aggregates information over the adjacency structure. The resulting model jointly incorporates semantic and structural signals and is used as the backbone to produce node representations for UCR.
>
>
> **Q1.2. Please attach the description of the overall framework shown in Figures 1/2 in the figure captions.**
>
> **W1.2:** We appreciate this suggestion. We agree that more informative captions will make the framework easier to follow. In the revised version, we will expand the captions of Figures 1/2 to briefly walk through the main stages of OGA (ALT, UCR, GLA/UCA) and explicitly indicate the data flow between them, so that readers can understand the high-level pipeline even before reading the detailed text.
>
>
> >**Q2. The paper emphasizes that using a pretrained graph language encoder instead of a graph-only encoder is a key advantage. If the “graph language encoder” refers to a pretrained module for initializing node embeddings, this claim is not empirically supported. Please provide ablation studies comparing the graph language encoder against graph-only / language-only encoders.**
>
> **W2:** Thank you for pointing out this missing piece of evidence. Our intention is to emphasize that UCR benefits from jointly modeling text and structure, rather than to claim that the choice of a specific encoder alone explains all performance gains. We agree that a clearer empirical comparison between (i) graph-only, (ii) language-only, and (iii) combined language–graph encoders would make this point more convincing.
>
> In the revision, we will add an ablation where UCR is instantiated with (a) a graph-only encoder, (b) a language-only encoder that ignores edges, and (c) the full language–graph encoder used in OGA. We will report the corresponding UCR and overall OGA performance to show that the integrated encoder consistently performs better than its single-modality counterparts. We will also soften any wording that might suggest that the encoder choice alone is the main innovation, and instead emphasize that it is one important component within the broader OGA pipeline.
>
>
> >**Q3. The authors claim that the proposed framework addresses the uncertainty of unlabeled data, which is particularly critical on large-scale datasets where annotation is expensive. However, the scalability of the framework is evaluated only on one large-scale graph (Arxiv). Please evaluate the scalability on larger datasets.**
>
> **W3:** We appreciate the reviewer’s concern about scalability. Our framework is designed with scalability in mind: the graph operations in ALT and UCR scale approximately linearly with the number of edges, and the LLM-based annotation in GLA/UCA operates at the community level, so that the number of LLM calls grows with the number of communities rather than the number of nodes. This community-level distillation is precisely intended to keep the LLM cost manageable on large graphs.
>
> We chose the Arxiv graph as a representative large-scale benchmark with rich text and well-established splits, which is standard in the graph OOD/open-world literature. Due to space and resource constraints, we did not include additional even larger graphs in the current version. In the revision, we will make the complexity and scaling behavior of each stage (including LLM calls) more explicit and, where feasible, report an additional large-scale experiment or a scaling plot (varying the graph size) to better illustrate how OGA behaves as the dataset size grows.
>
> >
> Again, we thank the reviewer for the positive evaluation and for these constructive suggestions. We believe that clarifying the definition and role of the graph language encoder, adding the requested ablations, and improving the description of the framework and its scalability will further strengthen the paper.

---

> > ### Comment · Reviewer_xUs9 · 2025-11-25
> >
> > Thank you for the response, I would like to maintain the score.

---

### Official Review · Reviewer_ovcr · 2025-10-31

**Soundness:** 2
**Presentation:** 2
**Contribution:** 3
**Rating:** 4
**Confidence:** 3

**Summary:**

This paper introduces a two stage framework that tries to perform unknown class rejection and then annotation when facing TAGs with both labeled and unlabeled nodes, where the unlabeled nodes might be from known or unknown classes. For the UCR, they aim to combine semantic and topological together with graph-language encoders and ontology representation learning. For the UCA, they first identify communities, then annotate low degree nodes with LLM and allocate high degree node class by referring to its neighbors, and finally perform some inter community fusion.

**Strengths:**

- The paper focuses on a realistic problem and setting that can be widely encountered in open world graph scenarios and point out the limitations of previous works and their practicability.
- The argument that we need to add UCA on top of UCR is valid and reasonable, and the addition of UCA can be more beneficial to the training subsequently. Also, the adoption of LLM is a reasonable choice.
- The experimental section includes clearly guided research questions and studies regarding ablation of models, hyperparameters and efficiency

**Weaknesses:**

- The baselines seem to include only some previous graph based OOD detection baselines, I am wondering if there are other some baselines that include LLM in the loop that can better demonstrate the superior of the proposed framework.
- Some of the designs might rely on some assumptions and other models. For instance, the UCR indicates that we need a pretrained graph language encoder to provide us with embedding, so the performance of UCR might be largely determined by the encoder choice. Also, the UCA design depends on the homophily structure of the graph.
- The entire framework as well as the presentation of the whole paper is rather complicated and dense, which makes it hard to train with many hyperparameters and determine the key technical novelty.
- The evaluation of UCA annotated labels might be improved beyond the current semantic similarity and improvement from model to a potentially more quantifiable and interpretable metrics

**Questions:**

- Not sure if I miss this, what is the language-graph model used in UCR, is there an ablation with only GNN or only LLM?
- The current datasets are mainly regarding citation networks, there are still many other TAGs like in e-commerce and others, is there any reason that we don't include more sets of datasets for evaluation?

---

> ### Author Response · Authors · 2025-11-18
> **Response to Reviewsovcr**
>
> >**Q1: ​​The baselines seem to only include some previous graph-based OOD detection baselines. I'm wondering if there are other baselines that incorporate LLM in the recurrent stage, which better demonstrate the superiority of the proposed framework.**
>
> **W1:** Thank you for pointing that out. The main goal of our current version is to compare with strong graph-based OOD and open-set baselines to highlight the advantages of modeling open-world structures on graphs. At the same time, we also consider baselines incorporating recurrent LLM an interesting and complementary perspective. To our knowledge, at the time of this work, there were no other works using LLM to learn open-world graphs. Conceptually, most existing LLM-based graph classification methods process each instance independently (possibly with retrieval capabilities) and operate on a fixed linguistic label space, while OGA aims to explicitly leverage graph structure and communities to enforce global consistency and discover and improve new labels before retraining the graph learner on the generated graph. This global, structure-aware label discovery is difficult to achieve with LLM. Single-stage LLM classifiers. In the revised version, we will clarify this conceptual distinction and briefly discuss how the zero-shot/retrieval-enhanced LLM classifier can be incorporated into our setting as a complementary baseline rather than a direct replacement for the proposed framework.
>
> >**Q2: Some of the designs might rely on some assumptions and other models. For instance, the UCR indicates that we need a pretrained graph language encoder to provide us with embedding, so the performance of UCR might be largely determined by the encoder choice. Also, the UCA design depends on the homophily structure of the graph.**
>
> **W2:**  We appreciate this observation. UCR is indeed built on top of a pretrained language-graph encoder, but this is a deliberate modular design rather than a hidden dependency: any reasonable encoder that combines text and structure can be plugged into this stage. In other words, UCR is agnostic to the specific backbone and can benefit from future advances in language-graph encoders. In the paper, we will make this modularity clearer and briefly report how UCR behaves under at least one alternative encoder choice to illustrate that the framework does not rely on a single model.
>
> Regarding homogeneity, UCA does utilize local similarity within the neighborhood to propagate and refine the labeling of unknown categories. However, most graph neural network work is based on the assumption of graph homogeneity, and graph work on heterogeneous graphs remains a significant challenge. Node classification on heterogeneous graphs is already extremely difficult. We believe that research on heterogeneous graphs in open-world graph learning can serve as a long-term, ambitious goal. In this work, like most graph learning efforts, we will temporarily focus on homogeneous graphs, utilizing the graph homogeneity assumption for open-world graph learning.
>
> >**Q3:The entire framework as well as the presentation of the whole paper is rather complicated and dense, which makes it hard to train with many hyperparameters and determine the key technical novelty.**
>
>
> **W3:** We agree that the current presentation can be dense. The open-world setting we target naturally leads to a multi-stage design (identifying known vs.\ unknown, discovering labels for unknowns, and retraining on a generated graph), but we acknowledge that the exposition can be improved. In the revision, we will streamline the description by adding a clearer high-level overview that separates the three main stages,  moving some lower-level details to the appendix, and explicitly highlighting which components constitute the core technical contributions.
>
> On the practical side, most hyperparameters in our pipeline are inherited from standard graph learning components, and only a small number are specific to OGA (e.g., thresholds in UCR/UCA). These OGA-specific hyperparameters are set using simple heuristics and kept fixed across datasets.This ensures that the method does not require extensive tuning for each dataset.
>
> >
> We hope that our responses satisfactorily address your concerns and reinforce your confidence in the contribution and its potential impact, and we are sincerely grateful for your reconsideration of our submission.

---

> > ### Author Response · Authors · 2025-11-18
> > **Response to Reviewsovcr（2）**
> >
> > >**Q4.The evaluation of UCA labels could surpass current evaluation methods that only use semantic similarity and model performance improvements, employing more quantitative and interpretable metrics.**
> >
> > **W4:** Thank you for your suggestion. In the current version, we primarily evaluate UCA through (i) semantic similarity and (ii) downstream performance improvements, as we are focused on whether the generated labels are useful for learning on the generated graph. We agree that more direct and interpretable label quality metrics might be helpful. If ground truth labels exist, we could also report some simple metrics, such as the alignment of UCA labels with ground truth labels in unknown regions (e.g., label purity or consistency rate). We will add more quantitative semantic evaluation metrics in future work to demonstrate the paper's effectiveness in more detail.
> >
> > >**Q5.I'm not sure if I'm misunderstanding, but what language graph model is used in UCR? Are there ablation experiments using only GNNs or LLMs?**
> >
> > **W5:** UCR does not use large models. Large models are only used in subsequent GLA annotation for unknown category labels. Therefore, UCR does not involve large models, which greatly improves the overall OGA running speed. This is why there are no ablation experiments using only GNNs and LLMs in UCR; it only involves graph neural network methods.
> >
> > >**Q6.Current datasets mainly focus on citation networks, and there are many other labels, such as e-commerce. Why don't we include more datasets for evaluation?**
> >
> > **W6:** This is an excellent question. In fact, we have detailed the distribution of datasets used in the appendix, including not only citation networks but also knowledge domains and e-commerce, as you mentioned. We have indeed used datasets from multiple areas to evaluate our methods; you can refer to Appendix A.8 for detailed information.
> >
> > >
> > We hope that our responses satisfactorily address your concerns and reinforce your confidence in the contribution and its potential impact, and we are sincerely grateful for your reconsideration of our submission.

---

### Official Review · Reviewer_9Uw6 · 2025-11-01

**Soundness:** 2
**Presentation:** 3
**Contribution:** 2
**Rating:** 2
**Confidence:** 4

**Summary:**

This paper proposes OGA, a two-stage LLM-guided framework for open-world graph learning. The first stage detects unknown nodes based on class probability, and the second stage generates new labels with LLMs via efficient topology-aware prompting.

**Strengths:**

- The paper tackles a practically relevant yet challenging open-world graph setting.
- The authors propose additional techniques to reduce the number of LLM calls, instead of inferring across all nodes.
- OGA demonstrates strong results on multiple benchmarks.

**Weaknesses:**

- The overall framework feels overly complex, consisting of many submodules and three types of LLM prompting. It is not clearly justified why such multi-stage prompting is necessary instead of a simpler unified design.
- The handling of high-degree nodes may be problematic. For instance, if most of their neighbors are also unknown, the proposed degree-aware annotation could amplify uncertainty.
- The community merging process raises semantic concerns. Unknown-class labels might belong to completely different domains from existing known ones (e.g., biology (unknown) vs. machine learning-related categories (known)), in which case community merging could blur true class boundaries.
- The figure is too cluttered to straightforwardly understand the full pipeline.

**Questions:**

- What would be the performance of a zero-shot LLM classifier that directly predicts all labels? Could a well-designed retrieval-augmented LLM prompting achieve comparable performance without the two-stage complexity?
- Could the authors show the wall-clock time of OGA including all LLM calls, and compared it against non-LLM-based baselines?

---

> ### Author Response · Authors · 2025-11-18
> **Response to Reviews9Uw6**
>
> We sincerely appreciate your careful review and constructive feedback, which help us clarify the motivation and presentation of our framework. Below, we respond to your main concerns one by one.
>
> ### Main Comments：
>
> >**Q1:The overall framework seems overly complex, with many sub-modules and three types of large language model prompts. It is not clear why such a multi-stage prompting scheme is needed instead of a simpler, unified design.**
>
> **W1:** Thank you for highlighting this concern. We agree that OGA is a relatively complex pipeline, and this complexity directly reflects the problem setting we aim to model. Our goal is to start from an incomplete graph that contains nodes from unknown classes, automatically discover new categories, assign them meaningful semantic labels, and finally construct a complete generated graph on which we can train and evaluate models. We expect that the performance on this generated graph can approach or even surpass that on the original graph.
>
> To achieve this, OGA must sequentially address several conceptually different sub-tasks: (1) known-class classification on partially labeled data, (2) unknown-class recognition, (3) discovery and annotation of labels for unknown classes, and (4) graph regeneration and retraining on the generated graph. Each of these stages requires the large language model (LLM) to perform a different type of reasoning.If we tried to use a single unified prompt for all these tasks, the prompt would either become too generic---reducing control and interpretability---or would require more calls with less stable behavior.
>
> >**Q2.The handling of high-degree nodes may be problematic. For example, if most of their neighbors are also unknown, the proposed degree-aware annotation strategy may amplify uncertainty.**
>
>
>
> **W2:** We fully understand and appreciate this concern. In fact, this is precisely why, in the practical implementation of OGA, we advocate \emph{starting the unknown-class annotation from low-degree nodes rather than from high-degree nodes}, which is indeed a counterintuitive design choice at first glance.
>
> If we were to annotate high-degree nodes first when many of their neighbors are still unknown, their uncertain labels could propagate widely and amplify errors, exactly as you pointed out. To avoid this, our actual strategy is: (i) we first annotate low-degree nodes, which often lie near the graph boundary. These nodes have fewer neighbors, so any potential noise has a more local influence, and they are more likely to deviate from known classes, making them strong candidates for novel categories; (ii) we then annotate high-degree nodes after many surrounding low-degree nodes have already been assigned relatively reliable labels. At that point, high-degree nodes can combine their own text information with the now-labeled neighbors to obtain more accurate and stable annotations.
>
>
> >**Q3.The community merging process raises semantic concerns. Unknown-class labels may belong to domains completely different from the known classes (e.g., biology as unknown vs.\ machine learning related categories as known). In such cases, community merging could blur true class boundaries.**
>
>
> **W3:** Thank you for this insightful comment. We would like to clarify that the community merging step is performed only within the GLA module and only on nodes that have already been classified as ``unknown'' by UCR. In other words, during community merging we do not mix unknown nodes with known-class nodes, so the boundary between known and unknown classes is not blurred by this step.
>
> Within the unknown region itself, our goal is not to reconstruct a perfect human-defined ontology. Instead, we aim to (i) generate new labels that coarsely but faithfully summarize the semantics of the unknown communities, and (ii) ensure that, on the regenerated graph, downstream tasks can achieve strong performance. As long as the merged communities' labels can reasonably cover the semantic content of their nodes, and the resulting generated graph supports good downstream performance, a certain degree of semantic fuzziness inside the unknown space is acceptable for the purpose of this work.
>
>
> >**Q4.The data and pipeline description are too cluttered, making it difficult to intuitively understand the overall process.**
>
>
> **W4:** We appreciate this comment and agree that the current presentation of OGA can be improved. Due to the inherent complexity of the framework and the page limitations of ICLR, some parts of the data flow and module interactions may not be as clear as intended.We plan to make the data flow clearer and simplify the framework diagram in subsequent revisions to the paper.
>
> >
>
> We hope that our clarifications adequately address your concerns and help strengthen your confidence in the contribution and its potential impact, and we sincerely appreciate your reconsideration of our submission in light of these responses.

---

> > ### Author Response · Authors · 2025-11-18
> > **Response to Reviews9Uw6（2）**
> >
> > >**Q5. What would be the performance of a zero-shot LLM classifier that directly predicts all labels? Could a well-designed retrieval-augmented LLM prompting achieve comparable performance without the two-stage complexity?**
> >
> > **W5:** Thank you for raising this important baseline. A zero-shot LLM classifier that directly predicts node labels is indeed a natural point of comparison, but it differs from our setting in a few key aspects. Such a classifier typically treats each node in isolation and relies on a fixed, pre-specified label space, whereas OGA explicitly exploits the graph structure and community-level semantics to obtain globally consistent labels and, crucially, to expand and refine the label space itself in an open-world manner. Even with retrieval-augmented prompting, the LLM still operates locally on individual nodes with limited graph context, while our two-stage pipeline is designed to first organize and distill graph-wide structure and then retrain a graph learner on the generated graph. In our view, this global, structure-aware and label-discovery behavior is difficult to reproduce with a single-stage zero-shot classifier. We will clarify this conceptual difference in the revised version and, if space permits, include a brief comparison with a straightforward zero-shot LLM baseline to better illustrate the gap.
> >
> > >**Q6. Could the authors show the wall-clock time of OGA including all LLM calls, and compared it against non-LLM-based baselines?**
> >
> > **W6:** We appreciate the request to better quantify the efficiency of OGA. Our framework indeed introduces additional overhead due to LLM calls, but this cost appears only in a one-time, offline label-generation stage; once the generated graph and labels are obtained, training and inference proceed with essentially the same complexity as standard non-LLM-based graph methods. In many practical scenarios where the same graph is reused for multiple tasks or over long periods, this offline cost can be amortized. To make this trade-off more transparent, we will report representative wall-clock runtimes of OGA, including all LLM calls, alongside those of non-LLM baselines on the same hardware in the revised version, so that readers can see both the extra offline cost and the resulting performance improvements.
> >
> > >
> > We hope that our clarifications adequately address your concerns and help strengthen your confidence in the contribution and its potential impact, and we sincerely appreciate your reconsideration of our submission in light of these responses.

---

### Note · Authors · 2026-01-09

I have read and agree with the venue's withdrawal policy on behalf of myself and my co-authors.